# DISTRICTNET: Decision-aware learning for geographical districting

**Cheikh Ahmed**
Polytechnique Montreal
cheikh-abdallahi.ahmed@polymtl.ca

**Alexandre Forel**
Polytechnique Montreal
alexandre.forel@polymtl.ca

**Axel Parmentier**
CERMICS, École des Ponts
axel.parmentier@enpc.fr

**Thibaut Vidal**
Polytechnique Montreal
thibaut.vidal@polymtl.ca

## Abstract

Districting is a complex combinatorial problem that consists in partitioning a geographical area into small districts. In logistics, it is a major strategic decision determining operating costs for several years. Solving districting problems using traditional methods is intractable even for small geographical areas and existing heuristics often provide sub-optimal results. We present a structured learning approach to find high-quality solutions to real-world districting problems in a few minutes. It is based on integrating a combinatorial optimization layer, the capacitated minimum spanning tree problem, into a graph neural network architecture. To train this pipeline in a decision-aware fashion, we show how to construct target solutions embedded in a suitable space and learn from target solutions. Experiments show that our approach outperforms existing methods as it can significantly reduce costs on real-world cities.

## 1 Introduction

Districting aims to partition a geographical area made of several basic units (BUs) into small, balanced, and connected areas known as districts. Districting has a wide array of applications in many areas such as electoral politics (Williams, 1995; Webster, 2013; Ricca et al., 2013), sales territory design (López-Pérez and Ríos-Mercado, 2013; Zoltners and Sinha, 2005), school zoning (Ferland and Guénette, 1990), and distribution (Zhou et al., 2002; Zhong et al., 2007). These problems are challenging because of the combinatorial complexity of assigning BUs to districts.

In this paper, we focus on districting and routing, one of the more complex forms of districting. In that case, the goal is to minimize the routing costs over all districts, where each district is serviced by a vehicle starting from a common depot. Since districting is a long-term strategic decision, the delivery requests in each district are unknown and modeled as a point process. As a two-stage stochastic and combinatorial problem, districting and routing can be solved to optimality only on very small instances. For example, in our experiments, finding the optimal solution of a districting problem with 60 BUs and 10 districts required around 400 CPU-core days. Further, while several heuristic methods have been proposed based on estimating district costs (Daganzo, 1984; Figliozzi, 2007), they tend to lead the search towards low-quality solutions on large instances (Ferraz et al., 2024).

In this study, we present a structured-learning approach called DISTRICTNET that integrates an optimization layer in a deep learning architecture, as shown in Figure 1. DISTRICTNET learns to approximate districting problems with a simpler one: the capacitated minimum spanning tree (CMST). DISTRICTNET predicts the cost of each arc of the CMST graph using a graph neural network (GNN)

38th Conference on Neural Information Processing Systems (NeurIPS 2024).

trained in a decision-aware fashion. This surrogate optimization model captures the structure of districting problems while being much more tractable. Using the CMST as a surrogate model is especially relevant because there is a surjection from the space of districting solutions to the space of CMST solutions. In other words, there always exists a CMST solution that is optimal for the original problem. The main challenge is to train a model that can find it.

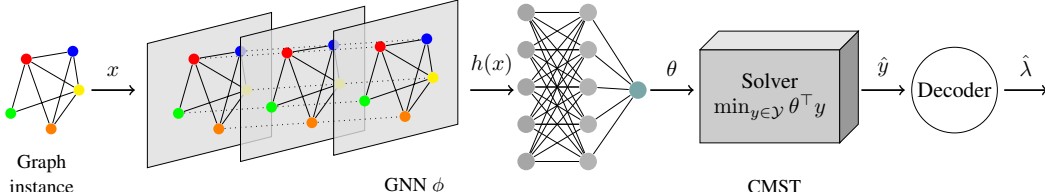

Figure 1: DISTRICTNET solves a complex districting problem by parameterizing and solving a CMST. The GNN $\phi$ predicts a vector of edge weights $\theta$ based on the covariates of the instance $x$. These edge weights parameterize a CMST, which is solved using a black-box combinatorial solver. The CMST solution $\hat{y}$ is finally converted into a districting solution $\hat{\lambda}$. Training this pipeline in a decision-aware manner requires propagating a loss gradient back to the GNN.

DISTRICTNET is trained to imitate a few optimal solutions obtained on small instances. We show that it leads to high-quality solutions for large districting problems and great out-of-distribution generalization. Further, our approach is robust to changes in the problem parameters such as a varying number of districts. This is valuable for practitioners, who usually evaluate a family of problems before picking a solution.

**Our contributions.** We present a structured learning approach to obtain high-quality solutions to large districting problems. The main component of our pipeline is a combinatorial optimization layer, a parameterized CMST, that acts as a surrogate to the original districting problem. We show how to learn from districting solutions by embedding CMST solutions into a suitable space and constructing target CMST solutions from districting ones. This allows training DISTRICTNET by imitation, minimizing a Fenchel-Young loss on a set of target solutions obtained on small instances. The value of our approach is demonstrated on real-world problems as DISTRICTNET generalizes to large problems and outperforms existing benchmarks by a large margin thanks to the combination of GNN and structured learning. While we focus our experiments on districting and routing, the method presented is not tailored to this specific application. DISTRICTNET is generic and could be applied to any geographical districting problem.

## 2 Problem statement

We model a geographical area as an undirected graph $G = (V, E)$, where $V$ is a set of vertices representing the BUs and $E$ is a set of edges representing the connections between them. A district $d \subset V$ is a subset of BUs. We say that a district $d$ is connected if its graph is connected. Let $\mathbb{1}(\cdot)$ be the indicator function that returns one if its argument is true and zero otherwise. We denote by $\mathcal{D}$ the set of connected districts and $N = |V|$ the number of BUs.

**Routing costs.** Let $d \in \mathcal{D}$ be a connected district. In districting and routing, the cost of a district is the expected distance of the smallest route that satisfies the demand requests. Hence, it is the expected cost of a traveling salesman problem $C_{\text{TSP}}(d) = \mathbb{E}_\xi[\text{TSP}(d, \xi)]$, where $\xi$ is the collection of demand requests in a district $d$.

Since the demand is not known a priori, it is modeled as a random variable. We assume that an average demand-generating distribution exists for each basic unit over the planning horizon so that $\xi_i \sim \mathfrak{D}_v, \forall v \in V$. Let $v_0$ be an additional vertex that represents the depot and is connected to all other nodes. Let also $\Pi(\xi)$ be the set of all routes that satisfy the demand requests of a district $d$ while starting and finishing at the depot, and denote by $\text{dist}(\pi)$ the total travel distance of a route $\pi \in \Pi$. Given a set of demand requests $\xi$, the minimum transportation costs are achieved by the route that visits all demand locations with minimum distance starting and ending at the depot so that

$$\text{TSP}(d, \xi) = \min_{\pi \in \Pi(\xi)} \text{dist}(\pi).$$

The district cost can be computed through a Monte Carlo evaluation by sampling a large set of demand scenarios and solving multiple independent TSPs. This evaluation is costly for large cities and districts. Therefore, evaluating the cost of a districting solution is hard even if districts can be evaluated in parallel.

**Districting.** The districting and routing problem minimizes the sum of all district costs. Let $C_{\text{TSP}}(d)$ denote the cost of a district $d$. A districting problem can be formulated in a general form as

$$\min_{\lambda \in \{0,1\}^{|\mathcal{D}|}} \quad \sum_{d \in \mathcal{D}} C_{\text{TSP}}(d)\lambda_d, \tag{1a}$$

$$\text{s.t.} \quad \sum_{d \in \mathcal{D}} \mathbb{1}(i \in d)\lambda_d = 1, \qquad \forall i \in V, \tag{1b}$$

$$\sum_{d \in \mathcal{D}} \lambda_d = k, \tag{1c}$$

where $\lambda_d$ is a binary variable that tracks whether a district is selected. Constraint (1b) states that each BU is selected in exactly one district of the solution. Constraint (1c) specifies that exactly $k$ districts are selected.

Additional constraints on the districts can be readily added to the problem. For instance, districting and routing typically aim to obtain districts close to a target size $t$ and include constraints on the minimum and maximum size of a district. Problem (1) can be formulated over a restricted set of districts $\mathcal{D}^r \subseteq \mathcal{D}$ that satisfy the minimum and maximum size constraints. Formally, it ensures that $\forall d \in \mathcal{D}^r, \underline{d} \leq |d| \leq \bar{d}$, where $(\underline{d}, \bar{d})$ are user-specified upper and lower bounds on the district size.

Although seemingly simple, Formulation (1) has an exponential number of variables. The number of feasible connected districts of size $t$ within a geographical area comprising $N$ BUs is on the order of $\mathcal{O}((e(\delta - 1))^{t-1} \cdot \frac{N}{t})$ where $\delta$ is the maximum number of neighbors for a given BU (Komusiewicz and Sommer, 2021). For instance, a city of $N = 120$ BUs and a target size of $t = 20$ with $\delta = 13$ has on the order of $10^{29}$ possible connected districts. The districting-and-routing problem has also been shown to be NP-hard by Ferraz et al. (2024). Hence, solving Problem (1) to optimality is only possible for small problem sizes.

**Existing approaches.** For real-world cities, evaluating a district's cost is too computationally demanding to be performed at each step of a search algorithm. Existing methods replace the cost $C_{\text{TSP}}(d)$ with a surrogate cost estimator. The defining aspect of existing methods is the specific model used to estimate the routing costs. Most approaches are based on the formula of Beardwood–Halton–Hammersley (BHH), which estimates the distance to connect randomly distributed points (Beardwood et al., 1959). This formula has been embedded into a hybrid search method combining a gradient method and a genetic algorithm (Novaes et al., 2000) and an adaptive large neighborhood search metaheuristic (Lei et al., 2015). Extensions of the BHH formula include Daganzo (1984), who adapted it for routing problems, and Figliozzi (2007) who extended it further for non-uniformly distributed demand requests. Recent works have shown that the BHH formula estimates stochastic TPS costs remarkably well empirically against more sophisticated regression functions for uniform distributions (see, e.g., Kou et al., 2022). Most closely related to our work, Ferraz et al. (2024) trained a GNN to predict district costs and embedded it in an iterative local search algorithm. They focused on solving single districting instances but did not investigate the generalization capabilities of their approach.

In contrast to the literature, we learn to approximate districting problems by using a surrogate optimization problem trained in a decision-aware fashion. An advantage of this framework is that it applies to a wide range of constrained partitioning problems: it can work with any general districting cost function $C(d)$. Hence, it could readily consider other metrics such as fairness, balancing and compactness of the district.

**Multi-instance learning and generalization.** Often, practitioners do not solve a single districting problem but study a family of problems with varying settings and parameters. Because there is a substantial effort needed to train a model, our goal is to obtain a pipeline able to solve multiple instances of districting. In our setting, generalizing over multiple instances means being able to generalize across:

 • *Cities*: each city has unique geographical and social characteristics, which affect the optimal districting solution. Our model should be able to identify the impact of these differences on districting and provide high-quality solutions for different cities.

- *Instance sizes*: the model should be able to handle instances with a different (typically larger) number of BUs than it was trained on. Since the training effort increases with the number of BUs, this allows solving large instances with a small training budget.
- *Problem parameters*: the model should be able to generalize to slight variations in the problem parameters, such as the minimum and maximum size of districts.

## 3  DISTRICTNET: From CMST to Districting

DISTRICTNET takes as input a labeled graph containing all the data of an instance. A graph neural network $\phi_w$ turns this labeled graph into edge weights. These weights parameterize a CMST instance, which is then solved using a dedicated algorithm. The CMST solution is then converted into a districting solution.

The first component of DISTRICTNET is a GNN $\phi_w : \mathbb{R}^{d_f} \to \mathbb{R}$ that assigns a weight to an edge depending on its feature vector. Applying this model to all edges of an instance returns a vector of edge weights $\theta \in \mathbb{R}^{|E|}$. Our model is made of two components: graph convolution layers that learn latent representations of graph edge features, and a deep neural network that converts these latent representations into edge weights.

The GNN is made of $L$ graph convolutional layers that update the features of edges through message passing. Following Morris et al. (2019), the update rule at the $l$-th layer is given by

$$h_e^{(l+1)} = \sigma\left(W_0^{(l)} h_e^{(l)} + \frac{1}{|\mathcal{N}(e)|} \sum_{j \in \mathcal{N}(e)} W_1^{(l)} h_j^{(l)}\right),$$ (2)

where $h_e^{(l)}$ is the feature vector of edge $e$ at the $l$-th layer, $\sigma$ is a non-linear activation function, $W_0$ and $W_1$ are learnable weight matrices, and $\mathcal{N}(e)$ is the set of neighboring edges of an edge $e$. Convolutional layers allow to capture the structure of the graph while being able to apply the prediction model to any graph size and connectivity structure. After the convolutional layers, a fully connected deep neural network transforms the latent representation produced by the GNN into an edge weight.

### 3.1  Combinatorial Optimization Layer

The second component of DISTRICTNET is the CMST layer parameterized by the vector of edge weights $\theta$. This layer acts as a surrogate for the original districting problem. The CMST is a well-studied graph optimization problem based on the minimum spanning tree problem. A minimum spanning tree is the subset of edges of a weighted graph that spans all vertices of the graph while minimizing the sum of the weight of its edges. Classically, the CMST extends this problem with the constraint that the number of vertices in each subtree does not exceed a predetermined capacity.

Let $\mathcal{T}$ be the set of connected subtrees with minimum and maximum size $(\underline{d}, \bar{d})$. We consider that an edge $e$ is in a subtree $s$ if both its extreme points are inside the subtree. The CMST problem with a target number of subtrees is then given by

$$\min_{\nu \in \{0,1\}^{|\mathcal{T}|}} \quad \sum_{s \in \mathcal{T}} \nu_s \sum_{e \in s} -\theta_e,$$ (3a)

$$\text{s.t.} \quad \sum_{s \in \mathcal{T}} \mathbb{1}(i \in s)\nu_s = 1, \qquad \forall i \in V,$$ (3b)

$$\sum_{s \in \mathcal{T}} \nu_s = k.$$ (3c)

Formulation (3) highlights the strong link between the CMST and the districting problem in (1). Both problems partition a graph into connected components with cardinality constraints. Any CMST solution can be converted into a districting solution by collecting all the nodes of a subtree into a district. Since several subtrees lead to the same district, this is a surjective mapping from the space of subtrees $\mathcal{T}$ to the space of districts $\mathcal{D}$. This also implies that, for any districting problem, there always exists a CMST problem such that their optimal solutions coincide.

**Solving the CMST.** Solving CMST problems is much more amenable than solving districting problems since it is a single-stage optimization problem with a linear objective. However, it remains

a combinatorial problem. Several methods have been proposed to solve it, ranging from expensive exact methods to quick heuristics.

DISTRICTNET is agnostic to the choice of CMST solver. In our experiments, we use the exact formulation in (3) to solve small instances to optimality, and we apply an iterated local search (ILS) heuristic for large instances. ILS alternates between two steps: (i) a local improvement step that guides to a local minimum, and (ii) a perturbation step to diversify the search. A key component of the algorithm is the initial solution. Randomly allocating BUs to districts is unlikely to return feasible solutions and, even when it does, it often leads to very poor solutions. Here, we use a modified Kruskal algorithm to exploit the structure of the CMST and quickly find a good initial solution. The details of our implementation of the ILS are given in Appendix A.

## 4  Training DISTRICTNET by imitation

DISTRICTNET is trained to imitate the solutions of a training set $\left\{x_i, \bar{\lambda}_i\right\}_{i=1}^n$. Each instance $x_i = (G_i, f_i)$ is a labeled graph with $G_i = (V_i, E_i)$ the instance graph and $f_i = \left\{f_i^e \in \mathbb{R}^{d_f}, \forall e \in E_i\right\}$ the set of edge feature vectors. A target districting solution $\bar{\lambda}_i$ is associated with each instance $x_i$. Since producing optimal or near-optimal districting problems is hard, building this training set is expensive. Thus, we train our model on a dataset of small instances with the hope that it generalizes well on large instances. Training DISTRICTNET by imitation amounts to solving

$$\min_w \sum_{i=1}^n \mathcal{L}\big(\phi_w(x_i), \bar{y}_i\big),$$

where $\mathcal{L} : (\theta, \bar{y}) \mapsto \mathcal{L}(\theta, \bar{y})$ is a loss function that quantifies the distance between a CMST solution corresponding to the edge weights $\theta$ with a target CMST solution $\bar{y}$. However, our training set does not contain any CMST targets but only districting targets. Training DISTRICTNET is thus achieved by three main steps: (i) introducing a new embedding of CMST solutions, (ii) converting districting targets $\bar{\lambda}_i$ into CMST targets $\bar{y}_i$, and (iii) defining a suitable loss function with desirable properties and deriving its gradient.

### 4.1  Embedding and target solutions

A CMST solution is entirely characterized by its edges. We can therefore build an embedding $\mathcal{Y}$ of the set $\mathcal{V}$ of solutions $\nu$ of the CMST problem given in (3) in $\mathbb{R}^{|E|}$ as $\mathcal{Y} = \big\{y(\nu) \colon \nu \in \mathcal{V}\big\}$ where $y(\nu) = \big(y_e(\nu)\big)_{e \in E}$ and $y_e(\nu) = \sum_{s \in \mathcal{T}} \nu_s \mathbb{1}(e \in s)$. We can then reformulate Problem (3) as

$$\max_{y \in \mathcal{Y}} \theta^\top y, \tag{4}$$

where, with a slight abuse of notation, we changed the problem from $\min$ to $\max$, which is without loss of generality. Since $\mathcal{Y}$ is finite, its convex hull $\mathcal{C}$ is a polytope. Therefore, the linear program given by $\max_{\mu \in \mathcal{C}} \theta^\top \mu$ is equivalent to Problem (4). The change of notation from a vertex $y$ to a moment $\mu$ emphasizes that $\mu$ takes values inside the convex hull $\mathcal{C}$. We now use the shorthand notation $\mu^*(\theta)$ to denote an optimal solution to $\operatorname{argmax}_{\mu \in \mathcal{C}} \theta^\top \mu$.

**From Districting to CMST.** As discussed previously, there is a surjection from the space of CMST solutions to the space of districting solutions. Given a target districting solution $\bar{\lambda}$, we denote by $\mathcal{Y}(\bar{\lambda})$ the set of feasible CMST solutions that lead to $\bar{\lambda}$.

To recover a target CMST solution, we introduce the constructor algorithm $\mathcal{A} : \lambda \mapsto y$ that maps a districting solution $\lambda$ to a CMST solution $y \in \mathcal{Y}(\lambda)$. This algorithm can be randomized. For instance, DISTRICTNET constructs districting solutions by solving a minimum spanning tree problem with random edge weights for each district $d \in \bar{\lambda}$. This can be efficiently done by applying Kruskal's algorithm in parallel for all the selected districts.

Finally, we define our target CMST solution $\bar{\mu} \in \mathcal{C}$ as

$$\bar{\mu} = \mathbb{E}[y|\bar{\lambda}, \mathcal{A}], \tag{5}$$

which is taken as an expectation over the realizations of the randomized algorithm $\mathcal{A}$. This allows us to convert our training set $\left\{x_i, \bar{\lambda}_i\right\}_{i=1}^n$ into $\left\{x_i, \bar{\mu}_i\right\}_{i=1}^n$ and enables training DISTRICTNET by imitation.

## 4.2 Fenchel-Young loss and stochastic gradient

Let $\bar{\mu}$ be a target CMST solution. We want DISTRICTNET to minimize the non-optimality of $\mu^*(\theta)$ compared to the target $\bar{\mu}$, that is, minimizing the loss $\max_{\mu \in \mathcal{C}} \theta^\top \mu - \theta^\top \bar{\mu}$. Minimizing this loss directly does not work because $\theta = 0$ is a trivial optimal solution. However, given a smooth strictly convex regularization function $\Omega(y)$, we can define the regularized problem (Blondel et al., 2020)

$$\max_{\mu \in \mathcal{C}} \theta^\top \mu - \Omega(\mu) \tag{6}$$

and the smoothed version of the non-optimality loss as:

$$\mathcal{L}_{\text{FY}}(\theta, \bar{\mu}) = \max_{\mu \in \mathcal{C}} \theta^\top \mu - \Omega(\mu) - (\theta^\top \bar{\mu} - \Omega(\bar{\mu})). \tag{7}$$

Denote by $\Omega^*(\theta)$ the Fenchel conjugate of $\Omega$, the regularized loss in Equation (7) is equal to $\mathcal{L}_{\text{FY}}(\theta, \bar{\mu}) = \Omega^*(\theta) + \Omega(\bar{\mu}) - \theta^\top \bar{\mu}$ (Dalle et al., 2022), which we recognize as the Fenchel-Young inequality (Blondel et al., 2020). Fenchel's duality theory then ensures that this loss has desirable properties. Notably, it is convex in $\theta$, non-negative, equal to 0 only if $\bar{y}$ is the optimal solution of (6), and its gradient can be expressed as $\nabla_\theta \mathcal{L}(\theta, \bar{\mu}) = \operatorname{argmax}_{\mu \in \mathcal{C}} \left( \theta^\top \mu - \Omega(\mu) \right) - \bar{\mu}$.

Practically, it remains to choose a suitable regularization function $\Omega$. When using a black-box oracle, a convenient choice that exploits the link between perturbation and regularization (Berthet et al., 2020; Dalle et al., 2022) is to define $\Omega(\mu)$ as the Fenchel dual of the perturbed objective $F(\theta) = \mathbb{E}\left[ \max_{\mu \in \mathcal{C}} (\theta + Z)^\top \mu \right]$, where $Z$ is a random variable with positive and differentiable density on $\mathbb{R}^{|E|}$. In that case, the gradient of the Fenchel Young loss with respect to $\theta$ can be computed as (Berthet et al., 2020)

$$\nabla_\theta \mathcal{L}_{\text{FY}}(\theta, \bar{\mu}) = \mathbb{E}_Z[\mu^*(\theta + Z)] - \bar{\mu}. \tag{8}$$

A stochastic gradient can therefore be conveniently computed using the Monte Carlo approximation $\frac{1}{M} \sum_{m=1}^{M} \operatorname{argmax}_{\mu \in \mathcal{C}} (\theta + Z_m)^\top \mu$ for the expectation, where $\{Z_m\}_{m=1}^{M}$ are sampled perturbations. If $\mathcal{A}$ is randomized, we also use a Monte Carlo approximation to estimate $\bar{\mu}$.

**Summary.** The novelty of our approach lies in our reconstruction of a CMST moment $\bar{\mu}$ from a districting solution $\bar{\lambda}$, which can be seen as a partially specified target $\bar{y}$. Alternatives in the literature generally consider completing the partially specified solution into the fully specified solution that minimizes the Fenchel Young loss (Cabannnes et al., 2020; Stewart et al., 2023). Our approach has the advantage of leading to the classic Fenchel Young loss, which is convex, whereas the infinum loss of Stewart et al. (2023) is only a difference of convex functions.

## 5  Numerical Study

We now evaluate the performance of DISTRICTNET on real-world districting and routing problems. We run repeated experiments and compare our approach to other learning-based benchmarks. We investigate the following aspects of DISTRICTNET: (i) its ability to generalize to large out-of-distribution instances from training on a few small instances, (ii) to variations in the instance parameters such as the district sizes, (iii) the role of the CMST surrogate model to allow this generalization.

Our experiments are implemented in Julia (Bezanson et al., 2017) except for the district evaluation methods, which are taken from Ferraz et al. (2024) and implemented in C++. All experiments are run on a computing grid. Each experiment is run on two cores of an AMD Rome 7532 with $2.4\,\text{GHz}$ and is allocated $16\,\text{GB}$ RAM. The code to reproduce all experiments presented in this paper is publicly available at https://github.com/cheikh025/DistrictNet under an MIT license.

### 5.1  Experimental Setting

Our instances include all the real-world cities in the United Kingdom used by Ferraz et al. (2024) and we extend it with additional cities in France. Hence, our test set contains seven real-world cities in the United Kingdom and France (Bristol, Leeds, London, Lyon, Manchester, Marseilles, and Paris), which contain between 120 and 983 BUs. Our goal is to provide high-quality solutions for several

values of the target district size $t$. This target size sets the bounds $(\underline{d}, \bar{d})$ on the district size $t \pm 20\%$ and the target number of districts as $k = \lfloor N/t \rfloor$.

**Features.** Each BU is summarized by a set of characteristics including its population, density, area, perimeter, compactness, and distance to the depot. For test instances, these summarizing statistics are taken from real-world data. The edge feature vector is constructed by averaging the feature vectors of the two BUs it connects. Additionally, we include the distance between the center of the two connecting BU into the edge feature vector. Thus, an instance is fully described by its geographical and population data.

**Training set.** To assemble a large and diverse training set, we generate new cities by perturbing real-world ones. First, we read the geographical data of 27 real-world cities in England (excluding the ones from the test set). From these initial cities, we generate $n = 100$ random connected subgraphs of size $N = 30$ BUs and sample the population of each BU according to a normal distribution $\mathcal{N}(8\,000, 2\,000)$ truncated between $5\,000$ and $20\,000$. For each instance $x_i$, we compute its optimal solution for the target size $t = 3$ by fully enumerating the possible districting solutions and evaluating their costs. This procedure generates our training set of $n = 100$ instances and associated solutions.

**Benchmarks.** We evaluate our method against four benchmark approaches that are based on learning estimators of district costs $C_{\text{TSP}}(d)$. They can be combined with any search method. In these experiments, they are also integrated within an ILS with a time limit of $20$ min.

We include (i) the method of Daganzo (1984) (BD), (ii) the method of Figliozzi (2007) (FIG), (iii) the method of Ferraz et al. (2024) (PREDGNN), and (iv) a deterministic approximation of the stochastic districting problem called AVGTSP. The first two approaches are extensions of BHH's formula and therefore are simple linear regression models. The third benchmark is based on training graph neural networks to estimate the district costs. Ferraz et al. (2024) train a GNN model to predict district costs for a fixed city and set of problem parameters and show that this approach outperforms BD and FIG. Since our focus is on generalization to multiple cities and problem parameters, our implementation extends the one of Ferraz et al. (2024): we train a single GNN using data from multiple small cities and evaluate its ability to generalize out-of-distribution. In contrast, AVGTSP estimates district costs by solving a TSP over the barycentres of all BUs within a district. This is a single-scenario approximation of the original stochastic districting problem that considers the expected demand realization in each BU.

The above benchmarks do not use a surrogate optimization model. Instead, they are trained in a traditional supervised learning fashion: to minimize the cost-estimation error on a training set of $10\,000$ districts and true costs taken from the same training instances as DISTRICTNET.

The details of our simulation setting, implementation, and additional experimental results are provided in the appendix.

## 5.2 Main results

We evaluate the ability of the different methods to generate good solutions on a diverse set of out-of-distribution instances. All methods are evaluated using the same performance metric: the total districting cost $C_{\text{TSP}}(d)$ of a districting solution as presented in Problem (1), where the expected costs are evaluated using a Monte Carlo approximation. We restrict the cities to $N = 120$ BUs and vary the target district size $t \in \{3, 6, 12, 20, 30\}$ for each of our seven test cities. This provides a set of 35 test instances, completely independent of the training data. The results are presented in Table 1 in the form of an ablation study. The table shows the relative difference in average costs achieved by each method on the test instances and the statistical significance is assessed using a one-sided Wilcoxon test. Example districting solutions are also given in Figure 2.

Table 1 shows that DISTRICTNET consistently outperforms the benchmarks as it produces districting solutions with significant cost reductions of around $10\%$ compared to all other methods. The benchmarks all provide similar performance, even PREDGNN, which uses a graph neural network but does not use structured learning. This highlights the ability of DISTRICTNET to generalize across various city structures and for larger instances thanks to the combination of a graph neural network and a differentiable optimization layer.

**Result 1.** DISTRICTNET *can solve a large family of districting problems with varying geographical and population data as well as varying problem parameters.*

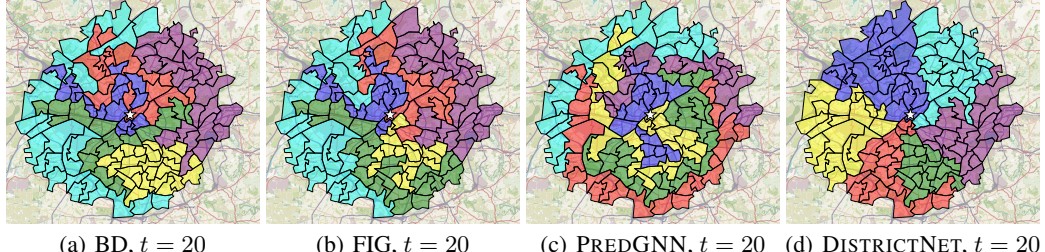

(a) BD, $t = 20$     (b) FIG, $t = 20$     (c) PREDGNN, $t = 20$   (d) DISTRICTNET, $t = 20$

Figure 2: Districting solutions given by BD, FIG, PredictGNN, and DistrictNet for the city of **Manchester** with district target sizes of 20 BUs. The depot is shown as a white star.

Table 1: Ablation study showing the value of combining GNN and structured learning.

|  | Average relative cost | $p$-value |
|---|---|---|
| Benchmark 1: BD, linear regression | 9.92 % | 4.9e-09 |
| Benchmark 2: FIG, linear regression | 10.01 % | 8.9e-09 |
| Benchmark 3: PREDGNN, unstructured learning with GNN | 11.91 % | 1.5e-10 |
| Benchmark 4: AVGTSP, no learning | 4.44 % | 2.7e-04 |
| DISTRICTNET: structured learning with CMST and GNN | 0.0 % | - |

**Large cities.** We perform an additional experiment to investigate the generalization to cities of large sizes. In Figure 3, we show the cost of the districting solution obtained with the benchmark methods relative to DISTRICTNET for varying city sizes. A value greater than 100% means that the benchmark performs worse than DISTRICTNET. We increase the number of BUs for each city and keep constant the target number of BUs in a district as $t = 20$. The time limit of ILS is kept to 20 min for all methods when $N < 400$ and increased to 60 min when $N \geq 400$.

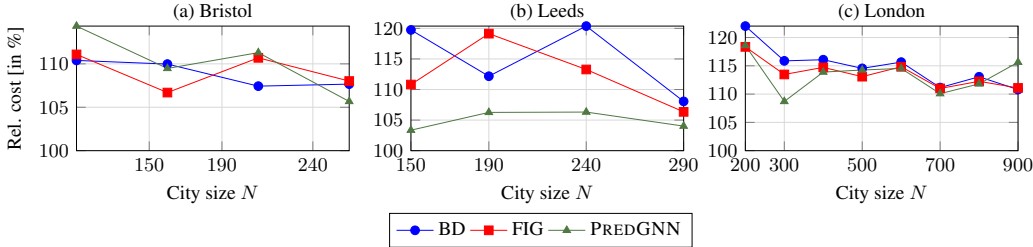

Figure 3: Cost relative to DISTRICTNET for target district size $t = 20$ and varying city size.

The results show that DISTRICTNET provides very good solutions up to the largest city sizes. It consistently outperforms the benchmarks even for large cities. These results are achieved despite DISTRICTNET being trained on small instances of size $N = 30$ BUs.

We investigate further the scalability of our approach by considering a large instance with 2,000 BUs in the Ile-de-France region. Each method is allowed 60 minutes to compute the districting solution, with the target district size set to 20 BUs. The results are presented in Table 2. DistrictNet provides the best performance, showing that it generalizes even to instances that are more than 60 times larger than the training ones.

**Result 2.** DISTRICTNET *provides high-quality solutions to even the largest real-world problems.*

**Why does unstructured learning fail?** One potential explanation for the poor performance of the benchmarks, in particular for PREDGNN, is the change in distribution between the training and test instances. As shown in Figure 4, the distribution of district costs varies greatly with the city and parameters. Since there is a shift in the data-generating distribution, the benchmarks, which ignore the structure of the districting problems, are not able to accurately predict the district costs resulting in poor overall performance.

Table 2: Comparison of districting costs for the Ile-de-France region with 2,000 BUs.

| Method | BD | FIG | PREDGNN | AVGTSP | DISTRICTNET |
|---|---|---|---|---|---|
| Absolute cost | 2379.0 | 2388.8 | 2295.2 | 2262.7 | **2205.7** |
| Cost relative to DISTRICTNET | +7.8% | +8.3% | +4.0% | +2.6% | **0.0%** |

**Result 3.** *Thanks to its surrogate optimization layer,* DISTRICTNET *captures the general structure of the districting problem. Thus, it is less sensitive to shifts in the data-generating distribution.*

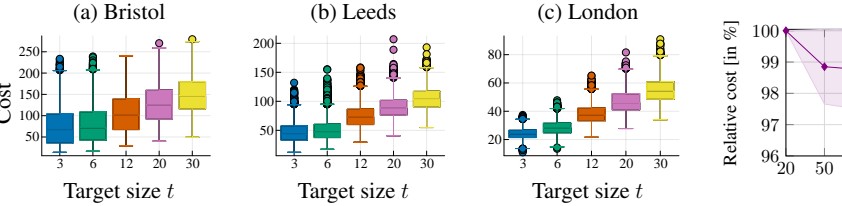

Figure 4: Distribution of district cost for varying target size.

Figure 5: Relative cost of DISTRICTNET with increasing data.

**The value of data for decision-aware learning.** Finally, we investigate the value of training data for DISTRICTNET. In Figure 5, we study the out-of-distribution performance of DISTRICTNET as the size of the training set increases. We show the average districting cost over all cities and target sizes relative to the cost for $n = 20$ and add a $95\%$ confidence interval as a shaded area. A value smaller than $100\%$ means that DISTRICTNET improves compared to its training with $n = 20$. The figure shows that DISTRICTNET can achieve low costs even with a surprisingly small number of training examples ($n = 50$). Increasing the number of examples tends to improve the results although with a diminishing return.

**Result 4.** DISTRICTNET *can be trained with a small computational budget and benefits from increasing the number of training examples.*

## 6 Related literature

In this work, we find near-optimal solutions to complex combinatorial problems by introducing a combinatorial layer in a deep neural network trained in a decision-aware fashion. Our work lies at the intersection of the learning-to-optimize literature and decision-aware learning.

**Learning to optimize.** Recent years have seen a significant increase in the use of machine learning for solving hard combinatorial optimization problems. Several graph-based learning approaches have been proposed for general combinatorial problems (Cappart et al., 2023). Deep reinforcement learning has been applied to solve typical combinatorial problems such as the traveling salesman and knapsack problems (Bello et al., 2016) and the minimum vertex cover and maximum cut problems (Dai et al., 2017). Gasse et al. (2019) presented a technique to improve the branch-and-bound process in mixed-integer linear programming by using graph convolutional neural networks.

The main advantage of learning-based methods is that, after a potentially expensive training procedure, they can generate good solutions to combinatorial problems in a short time. This has been shown to be effective in solving routing problems, which are complex combinatorial problems. Joshi et al. (2019) applied a beam search to solve the Euclidean TSP using GNNs and show notable improvements in solution quality, speed, and efficiency. Kool et al. (2019) combined a greedy rollout baseline and a deep attention model to solve several challenging routing problems such as the orienteering problem and the prize-collecting TSP.

**Decision-aware learning.** Decision-aware learning, on the other hand, looks into including an optimization layer in deep learning architectures. A key challenge in this area is to propagate a meaningful gradient through this non-smooth layer. Amos and Kolter (2017) developed a method to compute the gradients of quadratic programs by differentiating the Karush–Kuhn–Tucker optimality conditions. In the case of (integer) linear programs, propagating this gradient can be performed for

instance by introducing a log-barrier term to the LP relaxation (Mandi and Guns, 2020), using a piecewise-linear interpolation technique (Vlastelica et al., 2019), or using perturbation (Berthet et al., 2020). We refer the interested reader to Mandi et al. (2023) and Sadana et al. (2023) for surveys on decision-aware learning and its generalization as contextual stochastic optimization. Decision-aware learning has diverse applications such as approximating hard optimization problems by learning linear surrogate models (Ferber et al., 2023; Dalle et al., 2022).

Decision-aware learning allows the integration of complex algorithms with combinatorial behavior into deep learning architectures. Wilder et al. (2019) learn to solve hard combinatorial problems by learning from incomplete graphs using a differentiable k-means clustering algorithm. Stewart et al. (2023) present a differentiable clustering approach with a partial cluster-connectivity matrix. Our paper differs in two significant ways. First, we use a surrogate model with substantially more structure than clustering since it includes constraints on the size of the districts. This raises computational challenges, since the surrogate model remains NP-hard, but allows significant benefits in the quality of solutions. Second, while we have "full" districting solutions available, i.e., we know exactly what nodes need to be in the same clusters for a given training example, we have no information on the corresponding CMST solution. This leads us to introduce a randomized target construction algorithm, which leads to a well-defined loss function with advantageous properties.

Our approach belongs to the research stream that learns to approximate hard problems by easier ones. Key advantages of these approaches include being efficient at inference time since most of the computation effort is shifted offline, and great performance on test instances if they remain close to the training distribution. There are also downsides. First, until now, there are no known worst-case theoretical guarantees on the quality of the solution. Second, learning may be intensive computationally, both when generating the training instances and in the training algorithm. We refer the reader to Aubin-Frankowski et al. (2024) for a general discussion of these aspects and an analytical characterization of generalization bounds.

# 7 Conclusion

This paper presented a general pipeline to learn to solve graph-partitioning problems. It integrates a CMST as a surrogate optimization layer, which allows it to capture efficiently the structure of graph partitioning while providing solutions in a short time. We demonstrate the value of our pipeline on a districting and routing application. We show that our method outperforms recent and traditional benchmarks and is able to generalize to out-of-distribution instances. Thus, it can be trained on a small set of examples and applied to a wide array of cities, instance sizes, and hyperparameters.

Future work could investigate alternative approaches to solve the CMST problem during training and testing, such as exact methods based on column generation, which would integrate seamlessly with the pipeline presented in this paper. A limitation of our approach is applying DISTRICTNET only to districting and routing. Other geographical partitioning problems such as designing voting or school districts could be considered in future research.

# Acknowledgements

This research was enabled by support provided by Calcul Québec and the Digital Research Alliance of Canada, as well as funding from the SCALE AI Chair in Data-Driven Supply Chains. The authors also acknowledge the support of IVADO and the Canada First Research Excellence Fund (Apogée/CFREF).

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

# A    Supplementary Material: Methods and Implementation

In this part, we provide details on the four methods considered in this paper and their implementations. We include:

- in Appendix A.1 an illustration of how DISTRICTNET converts a districting problem into a CMST problem, and the training algorithm of DISTRICTNET,
- in Appendix A.2 the pseudo-code algorithm of ILS and its main components, and
- in Appendix A.3 background details on our benchmarks used in this paper as well as the architecture and hyperparameters used in our experiments.

## A.1    Additional Details on DISTRICTNET and Implementation

First, we show an illustrative example to highlight the similarity between districting problems and CMST problems. Then, we provide the algorithms used to train DISTRICTNET.

### A.1.1    Illustrative Example

We illustrate the link between districting problems and CMST problems in Figure 6. First, we show a simple districting instance with $N = 7$ BUs with the source node shown as a yellow star. In Figure 6 (b), we show the corresponding CMST instance that would be obtained when applying DISTRICTNET to predict the edge costs. The width of each edge is shown proportional to the edge's weight. In Figure 6 (c) we show the solution of this CMST instance. The solution has two subtrees stemming from the source node. Each subtree corresponds to a district.

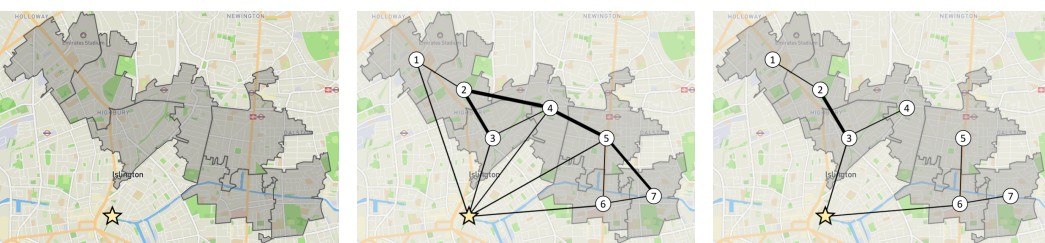

(a) Districting instance with $N = 7$ BUs

(b) CMST instance with edge width shown proportional to their weights

(c) Optimal CMST solution for $t = 3$

Figure 6: Small (a) districting problem converted into (b) a CMST problem with edge weights and (c) its corresponding solution: $d_1 = \{1, 2, 3, 4\}$ and $d_2 = \{5, 6, 7\}$.

### A.1.2    Training Algorithm

DISTRICTNET is trained following Algorithm 1. The hyperparameters of DISTRICTNET include the typical ones used in deep learning (number of epochs, batch size, learning rate) as well as the hyperparameters of the perturbation (number of samples $M$ and temperature $\varepsilon$). We note again that, in our implementation, DISTRICTNET uses the exact approach to solve each perturbed CMST, which is based on enumerating all the possible districting solutions and solving the exact formulation given in (3).

**Algorithm 1** Training DISTRICTNET

---

**Require:** Dataset $\mathcal{D} = \{x_i, y_i\}_{i=1}^n$
1: Initialize GNN model $\phi$
2: **for** $e = 1$ **to** `max_epochs` **do**
3:    **for all** $b = 1$ **to** `nb_batches` **do**
4:       Get batch $\mathcal{B}$ of training examples from $\mathcal{D}$
5:       **for all** $(x, y)$ in $\mathcal{B}$ **do**
6:          Predict edge costs $\theta \leftarrow \phi(x)$
7:          Sample perturbations $Z^{(1)}, \dots, Z^{(M)} \sim \mathcal{N}(0, 1)$
8:          **for** $m = 1$ **to** $M$ **do**
9:             Perturb edge costs $\theta^{(m)} \leftarrow \theta + \varepsilon Z^{(m)}$
10:           Solve perturbed CMST to obtain $y^{(m)}$
11:          **end for**
12:          Average perturbed solutions $\hat{y} \leftarrow \frac{1}{M} \sum_{m=1}^M y^{(m)}$
13:          Evaluate gradient of FY loss $\nabla_\theta \mathcal{L}_{\text{FY}}(y, \phi(x))$
14:          Make gradient step on parameters of $\phi$
15:       **end for**
16:    **end for**
17: **end for**
18: **Return** Trained model $\phi$

---

### A.2 Additional Details on Iterated Local Search (ILS)

All districting and CMST problems on large instances are solved using ILS. The detailed algorithm is given in Algorithm 2. ILS iteratively applies two methods: a local search that guides a solution to a local optimum, and a perturbation that explores the solution space. The local search algorithm is given in pseudocode in Algorithm 3. The perturbation algorithm is the same as the local search one except that each possible move is implemented with a given probability even if it does not improve the solution. This probability is typically very low. In our experiments, after hyperparameter tuning, we find that a probability of $1.5\%$ works well in most instances, which is consistent with the results of Ferraz et al. (2024).

---

**Algorithm 2** Iterated local search

**Input:** Initial districting solution $S_0$
**Initialize:** Best solution $S_{best} \leftarrow S_0$
**while** stopping criterion not met **do**
   Apply local search to $S$: $S' \leftarrow LS(S)$
   **if** $\text{cost}(S') < \text{cost}(S_{best})$ **then**
      Store best solution: $S_{best} \leftarrow S'$
   **end if**
   Apply perturbation to $S'$: $S \leftarrow P(S')$
**end while**
**Return:** $S_{best}$

---

**Algorithm 3** Local search: $LS$

**Input:** Current districting solution $S$
**while** improvement is found **do**
  **Initialize:** Best move $m_{best} = \varnothing$
  **for** each pair of districts $(D_a, D_b) \in S$ in random order **do**
    Identify border nodes between $D_a$ and $D_b$
    **for** each border node $i$ in $D_a$ **do**
      Evaluate all possible moves: move $i$ to $D_b$ or swap with a node $j$ in $D_b$
      Compare moves to $m_{best}$ and keep best
    **end for**
    **if** $m_{best}$ improves the solution **then**
      Apply best move $S \leftarrow m_{best}(S)$
    **end if**
  **end for**
**end while**
**Return:** $S$

---

#### A.2.1 Initial Solution

A feasible districting or CMST solution is a set of connected BUs of size $k$, where each subset has a number of BUs within the minimum and maximum acceptable size. Obtaining a feasible solution that satisfies these constraints is already NP-hard (Dyer and Frieze, 1985) and we observe that the

flow formulation proposed by Ferraz et al. (2024) does not scale well to large instances. Hence, we develop the following heuristic.

Given a set of edge weights, we first use the modified Kruskal algorithm given in Algorithm 4. This algorithm starts with all nodes in their own cluster and sorts all edge weights in increasing order. Then, it greedily merges clusters on the extreme points of the edges with lowest cost if the size of the merged cluster is below the maximum size. This provides a first solution that may still have too many clusters. In that case, we run the greedy merging algorithm given in Algorithm 5. This algorithm further reduces the number of clusters by continuously merging the two neighboring clusters that have the smallest combined size until the number of clusters meets the desired target $k$.

---

**Algorithm 4** Modified Kruskal

Initialize a cluster for each vertex $u \in G$
Sort edges of $G$ by increasing weight as $E_{sorted}$
**for** each edge $(u, v)$ in $E_{sorted}$ **do**
  **if** vertices $u$ and $v$ are in different clusters and $|u| + |v| \leq t$ **then**
    Merge clusters of $u$ and $v$
  **end if**
**end for**

---

**Algorithm 5** Greedy merging

**while** number of districts is larger than target $k$ **do**
  Find neighboring districts $D_a$ and $D_b$ with the smallest combined size
  Merge $D_a$ and $D_b$
**end while**

---

However, the initial solutions obtained may not always meet the limit size requirements. To address this, we have incorporated a penalty function in the local search algorithm, which penalizes clusters not conforming to size limits, thereby guiding the local search towards feasible solutions. In our experiments, a feasible solution is thus almost always found at the first iteration of the ILS algorithm.

Empirically, this method efficiently finds feasible initial solutions in our main experiments. However, they are not sufficient for the largest instances (when $N \geq 600$ in our experiments). Thus, we introduce an additional repair algorithm given in Algorithm 6, which is used only in the experiment presented in Figure 3. This repair algorithm adjusts each district to meet the specified minimum and maximum size constraints by adding or removing nodes from neighboring districts while maintaining overall connectivity.

---

**Algorithm 6** Repair

**Require:** Current districting solution $S$
Sort districts in $S$ by size in increasing order
**for** each district $d \in S$ **do**
  **if** $|d| < \underline{d}$ **then**
    Add nodes from neighboring districts to $D$
  **end if**
**end for**
Sort districts in $S$ by size in decreasing order
**for** each district $d \in S$ **do**
  **if** $|d| > \bar{d}$ **then**
    Remove nodes from $d$ and add them to neighboring districts
  **end if**
**end for**

---

### A.3 Benchmark Methods

**BD.** Following Daganzo (1984), BD estimates the cost of a district as

$$\text{dist}_{\text{BD}}(d) = \beta \sqrt{A_d R_d} + 2\Delta_d, \tag{9}$$

where $A_d = \sum_{i \in d} a_i$ represents the total area of the district with $a_i$ being the area of the $i-$th BU, and $R_d$ denotes the expected number of demand requests within the district. The term $\Delta_d$ is the average distance between the depot and a demand point. Because we do not make assumptions on the shape of the BUs, we compute this term using a Monte Carlo approximation. The parameter $\beta$ is a

parameter that adjusts the influence of area and demand on the cost. Hence, BD is a linear regression and training it amounts to finding the right value for $\beta$.

**FIG.** Following Figliozzi (2007), the cost of a district is given by

$$\text{dist}_{\text{FIG}}(d) = \beta_1 \sqrt{A_d R_d} + \beta_2 \Delta_d + \beta_3 \sqrt{\frac{A_d}{R_d}} + \beta_4, \tag{10}$$

where $\beta = (\beta_1, \beta_2, \beta_3, \beta_4)$ now represents a vector of parameters that modulates the cost estimation. Clearly, FIG is an extension of BD and we train it in the same way.

**PREDGNN.** Following Ferraz et al. (2024), PREDGNN uses a GNN to estimate the cost of a district, i.e., $\text{dist}_{\text{PREDGNN}}(d) = \Phi(G_d)$, where $\Phi$ is the trained GNN and $G_d$ is a district graph. The district graph is a labeled graph where each node $i \in d$ is described by the features

$$f_{i,d} = (p_i, \sqrt{p_i}, a_i, \sqrt{a_i}, q_i, \rho_i, \delta_i, e_{i,d}) \tag{11}$$

where $p_i$ is the population of the BU $i$, $q_i$ its perimeter, $a_i$ its area, $\rho_i$ its density, $\delta_i$ its distance to the depot, and $e_{id}$ is an inclusion variable that takes the value 1 if BU $i$ belongs to the district $d$ and 0 otherwise. The GNN processes $G_d$ to learn a latent representation that captures the structure of the district. This is done in two phases. First, a message-passing algorithm is applied at the node level, then an aggregation layer summarizes the whole graph into a single latent vector. This latent graph representation is then fed to a feedforward NN to predict the district cost. This architecture is summarized in Figure 7.

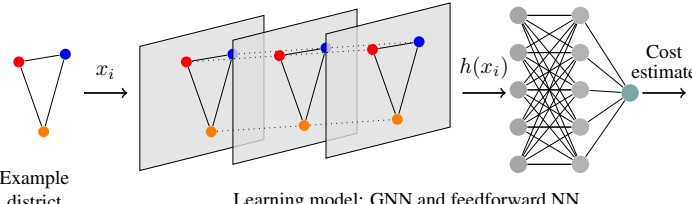

Figure 7: PREDGNN estimates the cost of a district using a GNN and a feedforward NN. First, the GNN applies a message-passing algorithm to capture the structure of the graph. Then, an aggregation layer provides the graph embedding. This is post-processed by the feedforward NN, which outputs a cost estimate.

**Key differences with DISTRICTNET.** PREDGNN and DISTRICTNET both use GNN to learn the structure of labeled graphs. PREDGNN uses node-based features available at the BU level. In contrast, DISTRICTNET uses edge-based features, averaging the attributes of the two BUs connected to each edge. Hence, while PREDGNN focuses on the properties of individual BUs, DISTRICTNET also captures the spatial relationships between connected BUs. Further, DISTRICTNET does not use the final aggregation layer for global graph embedding of PREDGNN since it is applied independently to each edge. This allows a finer-grained representation of the graph.

**AVGTSP.** This benchmark replaces the districting-and-routing problem in Equation (1) by the surrogate optimization problem given by

$$\min_{\lambda \in \{0,1\}^{|\mathcal{D}|}} \sum_{d \in \mathcal{D}} \text{TSP}(d, \mathbb{E}_\xi[\xi]) \lambda_d, \tag{12}$$

that is, it exchanges the $\min$ and expectation operations in the objective function of the original problem. This yields a deterministic problem, where the expected districting cost is approximated by the cost of the expected demand in each BUs. In many stochastic problems, this approach often leads to poor results (see e.g., Chapter 4.2 of Birge and Louveaux (2011)). This is because this approximation ignores the variance of the random variable $\xi$.

Still, we implement this method as a non-learning-based baseline. In our setting, the demand distribution is known and the expected demand in each BU can be easily computed. For each BU, the average demand request $E_\xi[\xi]$ is a single request that appears at the barycentre of the area. Evaluating a district cost thus reduces to solving a TSP over the barycentres of all its BUs. As for all other methods, we integrate this approximation in an iterated local search.

### A.4 Architectures and Hyperparameters

Several hyperparameters control the different methods used in this paper. We give here the architectures and hyperparameters used in all our experiments. We also provide a summary of the computational effort for training the model in Table 3. The table illustrates well the two main sources of computational efforts for the GNN-based methods. The training time of PREDGNN is relatively longer because it uses a large number of district costs examples. On the other hand, DISTRICTNET needs to evaluate more districts to find the optimal solutions of its 100 training instances but is quick to train. This can be seen in the second column of Table 3 that shows the number of districts whose cost is evaluated to compute the optimal solution of $n = 100$ instances. Notably, the district cost evaluation can all be performed in parallel.

Table 3: Summary of parameters and computational effort for training the different methods.

|  | Nb. training examples $n$ | Nb. district cost evaluation | Training Time |
|---|---|---|---|
| BD | $10^4$ | $10^4$ | 8 seconds |
| FIG | $10^4$ | $10^4$ | 3 seconds |
| PREDGNN | $10^4$ | $10^4$ | 16 hours |
| DISTRICTNET | $10^2$ | $62 \cdot 10^4$ | 38 minutes |

**Architecture.** The GNN of DISTRICTNET uses three graph convolution layers, each with a hidden size of 64 and Leaky ReLU activation functions. The DNN section uses three dense layers: two layers mapping with 64 inputs to 64 outputs and then one layer with an output of size 32. All three layers use Leaky ReLU activations. Finally, the final layer converts the latent 32-dimension vector into a one-dimensional output.

For PREDGNN, we maintain the structure proposed by Ferraz et al. (2024), with the exception that we replace the Structure2vec layers with GraphConv layers. The update rule for the GraphConv layers is $x_i^{t+1} = \text{ReLU}(W_1 x_i^t + \sum_{j \in N(i)} W_2 x_j^t)$, where $W_1$ and $W_2$ are the weight matrices and $N(i)$ is the set of neighbors for node $i$. In contrast, the Structure2vec update rule applied in the original model is $x_i^{t+1} = \text{ReLU}(W_1 x_i^t + W_2 \sum_{j \in N(i)} x_j^t)$. This is a minor modification and should not alter its performance, as also stated by Ferraz et al. (2024).

The architecture of PREDGNN thus consists of four GraphConv layers. Each layer has a hidden size of 64 units and uses ReLU activation functions. The final layer produces an output of size 1028. We then aggregate these node embeddings to form a global graph embedding. This global embedding is initially processed by a feedforward layer, resulting in a 100-dimensional vector. A second and final layer reduces this to a single output, representing the cost of the district.

**Hyperparameters.** The only hyperparameter of BD and FIG is the number of samples used in the Monte Carlo approximation of the average distance to requests. We use 100 scenarios in all our experiments.

PREDGNN uses a batch size of $|\mathcal{B}| = 64$, a learning rate of $1e^{-4}$, and train the model for $10^4$ epochs. We further set a time limit of 24 hours for training and stop the process early if no significant change of the loss (i.e., greater than $1e^{-4}$) is observed in the last $1\,000$ epochs.

DISTRICTNET uses a batch size of $|\mathcal{B}| = 1$ and a learning rate with initial value of $1e^{-3}$ with a decay rate of 0.9 applied every 10 epochs and a minimum rate of $1e^{-4}$. The model is trained for 100 epochs. The target CMST solution in Equation (5) is constructed using a single observation of our random constructor (Kruskal with random weights). The perturbation $Z$ is set to a multivariate standard Gaussian and we $M = 20$ samples to approximate the expected gradient in Equation (8). The randomized target constructor uses $1\,000$ samples.

# B Supplementary Material: Data Collection and Generation

In this part, we describe how to collect and generate the real-world data used in all experiments. First, we describe the test instances, the four large cities on which we evaluate all the methods. Then, we present how we use real-world data to generate a set of training instances that can be arbitrarily large. We also discuss the distributional assumptions used to simulate the random demand over a city.

## B.1 Test Instances: Real-World Cities and Population

We use the four cities of Bristol, Leeds, London, and Manchester for our test instances. A summary of these instances is given in Table 4. The table shows the statistics on the population, area, and density of the BUs composing the four test cities. It shows that, while the area and density may vary across cities, the population statistics are relatively constant. This is not surprising since BUs tend to be designed to have similar populations. The geographical data including the boundaries of each city and BUs can be accessed at Uber Movement: https://movement.uber.com/. Additionally, census data is available at the Office for National Statistics (ONS) website: ONS. For the French cities, geographical data including the boundaries were obtained from Opendatasoft. Population data for these regions was sourced from the National Institute of Statistics and Economic Studies (INSEE) available at INSEE.

Table 4: Statistics on the BUs of the seven test cities, adapted from Ferraz et al. (2024)

|  |  | Bristol | Leeds | London | Manchester | Paris | Lyon | Marseilles |
|---|---|---|---|---|---|---|---|---|
| Population (thousands) | Average | 8.28 | 7.76 | 9.01 | 8.21 | 2.49 | 2.71 | 2.29 |
|  | Std. | 2.13 | 1.80 | 1.90 | 2.12 | 0.74 | 1.37 | 0.77 |
|  | Minimum | 5.23 | 5.20 | 5.43 | 4.96 | 0.36 | 0.28 | 0.58 |
|  | Median | 7.85 | 7.40 | 8.69 | 7.76 | 2.33 | 2.46 | 2.27 |
|  | Maximum | 18.16 | 16.17 | 24.97 | 15.87 | 4.83 | 6.47 | 4.88 |
| Area $(km^2)$ | Average | 16.39 | 6.80 | 1.62 | 3.05 | 0.093 | 6.77 | 0.52 |
|  | Std. | 30.16 | 10.40 | 1.88 | 3.60 | 0.07 | 7.62 | 0.94 |
|  | Minimum | 0.63 | 0.35 | 0.30 | 0.59 | 0.018 | 0.11 | 0.07 |
|  | Median | 2.80 | 3.05 | 1.16 | 2.13 | 0.07 | 4.21 | 0.29 |
|  | Maximum | 171.21 | 94.13 | 22.30 | 35.69 | 0.45 | 38.96 | 6.81 |
| Density | Average | 3.12 | 2.99 | 8.92 | 4.01 | 34.3 | 1.94 | 9.59 |
|  | Std. | 2.62 | 2.80 | 5.25 | 2.32 | 16.5 | 2.649 | 6.98 |
|  | Minimum | 0.05 | 0.10 | 0.36 | 0.14 | 2.6 | 0.0039 | 85.18 |
|  | Median | 2.89 | 2.41 | 7.77 | 3.79 | 34.73 | 0.71 | 7.5 |
|  | Maximum | 12.72 | 25.20 | 28.27 | 16.36 | 129.09 | 11.45 | 32.6 |

## B.2 Training Instances

For our experiments, we utilize real-world data from a diverse set of 27 cities shown in Table 5. Apart from two outliers, these cities are smaller than our test instances, with around 30 to 50 BUs. While we read the true boundaries of the cities, we generate their population randomly. We use a normal distribution $\mathcal{N}(8\,000, 2\,000)$ truncated within the range $[5\,000, 20\,000]$ to be close to the true population distribution of BUs.

To generate the city graphs used in the training of DISTRICTNET, we sample randomly connected subgraphs from the train cities. The training instance generation process is given in pseudocode in Algorithm 7, and the subgraph sampling algorithm is given in Algorithm 8. Finally, an artificial central depot is placed at the centroid of the resultant polygon. This procedure allows us to create a training set of arbitrary size that contains realistic (but small-sized) training instances. Note also that there is no contamination between the training and test instances.

Table 5: Cities used to generate training instances and corresponding number of BUs

| City | N | City | N | City | N | City | N |
|---|---|---|---|---|---|---|---|
| Barnet | 41 | Coventry | 42 | Leicester | 37 | | |
| Birmingham | 132 | Derby | 31 | Liverpool | 61 | Sefton | 38 |
| Bolton | 35 | Doncaster | 39 | Newham | 37 | Sheffield | 70 |
| Bradford | 61 | Ealing | 39 | Nottingham | 38 | Southwark | 33 |
| Brent | 34 | Greenwich | 33 | Oldham | 33 | Stoke | 34 |
| Brighton | 33 | Kirklees | 59 | Plymouth | 32 | West Midlands | 684 |
| Cardiff | 48 | Lambeth | 35 | Rotherham | 33 | Wigan | 40 |

---

**Algorithm 7** Training instance generation

---

**Input:** Number of instances $n$, real-world training cities
**for** $i = 1$ to $n$ **do**
    Sample a city from the train cities with probability proportional to its size
    Sample a connected subgraph of this city of size $N = 30$ BUs
    Sample the population of each BU from truncated normal distribution
    Place central depot
**end for**
**Return:** Set of $n$ training instances

---

### B.3 District Demand and District Cost

We suppose that the demand within each BU follows a fixed distribution. In each period (e.g., a day), a number of demand requests in each BU. The number of demand requests follows a Poisson distribution with the rate being proportional to the population density of the BU, expressed as $n \times \kappa$. Here, $n$ represents the population count of the BU, and $\kappa$ is set to $96/(8\,000 \times t)$. This value of $\kappa$ is selected to realistically approximate a scenario where a district typically handles around a hundred stops. Each demand request is located randomly with a uniform distribution over the geographic location covered by the BU.

Evaluating the cost of a district requires solving a stochastic TSP since $C_{\text{TSP}}(d) = \mathbb{E}_{\xi}[\text{TSP}(d, \xi)]$. This is done through a Monte Carlo estimation. To accurately estimate the operational costs of our districting solutions, we sample 100 demand scenarios for each basic unit. For each district, we collect the demand requests from the BUs it contains and solve 100 independent TSP problems. We then evaluate a district's expected operational cost by averaging the calculated TSP costs.

Each TSP is solved using the Lin-Kernighan-Helsgaun (LKH) algorithm (Lin and Kernighan, 1973; Helsgaun, 2017). We use the publically available implementation from: http://akira.ruc.dk/~keld/research/LKH-3/. This heuristic is widely accepted as a state-of-the-art heuristic for TSPs. It is based on a local search strategy that consistently returns near-optimal solutions.

---

**Algorithm 8** Sampling a connected subgraph of size $N$

---

**Input:** Graph $G$, maximum size $N$
Select a random starting node $u$ from $G$ and set $s = \{u\}$
**while** $|s| \leq N$ **do**
    Build $\mathcal{N}(s)$, the set of all BUs that are neighbors of $s$
    Randomly select a node $v \in \mathcal{N}(s)$ and add it to $s$
**end while**
**Return:** Connected subgraph of $G$

---

# C   Supplementary Material: Additional Results

In this part, we provide additional experimental results. We give:
- an overview of results on small training and validation instances in Appendix C.1,
- a detailed table presenting the individual results of experiments in Appendix C.2,
- a table presenting the compactness of the districts obtained in Appendix C.3, and
- examples districting solutions for all methods on various instances in Appendix C.4.

## C.1   Out-of-Sample but In-Distribution

We perform a small experiment to investigate the in-distribution generalization ability of the different methods. That is, we study the setting where the training and test instances are sampled from the same distribution. We generate 200 instances following the procedure described in Appendix B.2 and split this into two sets of 100 instances. The first set is used to train all the methods and the second is used to evaluate them out-of-sample.

All instances are of size $N = 30$ and with a target district size $t = 3$. Hence, we can solve them to optimality in a reasonable time using a full enumeration of the possible districts and the exact formulation given in Problem (1). We solve to optimality all the districting problems of BD, FIG and PREDGNN and all the CMST problems of DISTRICTNET during both training and testing. Further, since we consider only small instances here, we can measure the optimality gap of all methods: the relative difference between the cost of the true optimal solution and the one of the methods.

The optimality gap of the four methods is shown in Table 6 for both the training and testing instances. The results show that DISTRICTNET achieves the smallest average test gap from the optimal solutions compared to other benchmarks. BD and FIG have similar performance, which can be expected since they estimate the district costs similarly. PREDGNN has the worst average performance, although less variations since it has a smaller maximum gap than BD and FIG.

Table 6: Gap to optimal districting solutions on train and test instances.

|  | Train | | | Test | | |
|---|---|---|---|---|---|---|
|  | Avg. | Max. | Min. | Avg. | Max. | Min. |
| BD | 4.0 % | 11.25 % | 0.27 % | 4.09 % | 11.06 % | 0.42 % |
| FIG | 4.24 % | 15.98 % | 0.27 % | 4.09 % | 12.16 % | 0.42 % |
| PREDGNN | 3.37 % | 10.05 % | 0.05 % | 4.64 % | 11.89 % | 0.22 % |
| DISTRICTNET | 1.86 % | 7.48 % | 0.0 % | 2.34 % | 5.52 % | 0.17 % |

## C.2   Detailed results for each city

In Table 7, we present the individual experiment results that have been summarized in the ablation study presented in Section 5. The table shows the districting cost achieved by the methods on each test city and for each target district size. The lowest cost is shown in blue and the second-best in orange. It highlights that DISTRICTNET provides the lowest count on 27 out of 35 test instances. Further, it shows that DISTRICTNET lead to significant savings, as it can reduce costs by up to 13% in the best case. Overall, the cost savings are higher for large target district sizes, and for cities in the United Kingdom that are closer to the training instances.

## C.3   District compactness

Compactness is an important criterion in districting problems as it often leads to efficient and fair divisions. Compactness here does not refer to the topological property. In the districting terminology, it refers to a shape property. We measure compactness using Reock's score, a commonly used metric measure defined as the ratio of the area of the district to the area of the minimum enclosing circle that contains the district (Reock, 1961). A higher score indicates greater compactness, with compactness equal to 1 when the district is circular. As shown in Table Table 8, DISTRICTNET consistently provides more compact districts compared to other methods, suggesting that it can design geographically tighter districts.

Table 7: Districting costs across different cities and target district sizes for our method and benchmarks. The best result is shown in blue and the second best in orange. The last column shows the relative difference between DISTRICTNET and the best or second-best method.

| City | $t$ | BD | FIG | PREDGNN | AVGTSP | DISTRICTNET | (Rel.) |
|------|-----|----|-----|---------|--------|-------------|--------|
| Bristol | 3 | 1944.28 | 1915.43 | 1967.8 | 2015.68 | 1873.07 | $(-\mathbf{2.2}\%)$ |
| | 6 | 1274.12 | 1308.06 | 1269.82 | 1325.37 | 1192.9 | $(-\mathbf{6.1}\%)$ |
| | 12 | 896.3 | 912.61 | 861.38 | 859.18 | 819.54 | $(-\mathbf{4.6}\%)$ |
| | 20 | 696.9 | 713.24 | 681.35 | 667.11 | 633.15 | $(-\mathbf{5.1}\%)$ |
| | 30 | 564.12 | 564.12 | 600.48 | 558.41 | 539.02 | $(-\mathbf{3.5}\%)$ |
| Leeds | 3 | 1438.21 | 1433.44 | 1498.61 | 1416.05 | 1400.64 | $(-\mathbf{1.1}\%)$ |
| | 6 | 933.17 | 953.35 | 957.82 | 920.49 | 901.2 | $(-\mathbf{2.1}\%)$ |
| | 12 | 677.64 | 670.99 | 653.65 | 682.18 | 586.7 | $(-\mathbf{10.2}\%)$ |
| | 20 | 531.27 | 540.59 | 483.95 | 461.18 | 442.97 | $(-\mathbf{3.9}\%)$ |
| | 30 | 463.06 | 497.07 | 413.63 | 416.14 | 389.66 | $(-\mathbf{5.8}\%)$ |
| London | 3 | 745.08 | 745.78 | 748.21 | 773.0 | 737.51 | $(-\mathbf{1.0}\%)$ |
| | 6 | 494.7 | 486.35 | 502.83 | 495.72 | 465.67 | $(-\mathbf{4.3}\%)$ |
| | 12 | 346.06 | 331.42 | 350.25 | 329.04 | 300.48 | $(-\mathbf{8.7}\%)$ |
| | 20 | 271.57 | 265.67 | 282.67 | 249.75 | 227.66 | $(-\mathbf{8.8}\%)$ |
| | 30 | 207.12 | 219.87 | 228.5 | 211.41 | 182.55 | $(-\mathbf{11.9}\%)$ |
| Lyon | 3 | 1466.5 | 1466.41 | 1539.63 | 1570.19 | 1446.49 | $(-\mathbf{1.4}\%)$ |
| | 6 | 918.66 | 912.29 | 953.14 | 938.32 | 884.61 | $(-\mathbf{3.0}\%)$ |
| | 12 | 626.17 | 627.82 | 649.68 | 614.44 | 585.75 | $(-\mathbf{4.7}\%)$ |
| | 20 | 554.02 | 549.88 | 595.85 | 500.44 | 471.62 | $(-\mathbf{5.8}\%)$ |
| | 30 | 549.62 | 517.66 | 495.97 | 494.76 | 432.99 | $(-\mathbf{12.5}\%)$ |
| Manch. | 3 | 1173.48 | 1162.94 | 1240.42 | 1158.66 | 1165.72 | $(0.6\%)$ |
| | 6 | 781.3 | 770.16 | 824.39 | 739.87 | 733.74 | $(-\mathbf{0.8}\%)$ |
| | 12 | 565.1 | 564.8 | 558.79 | 546.07 | 474.76 | $(-\mathbf{13.1}\%)$ |
| | 20 | 428.34 | 440.67 | 469.02 | 360.78 | 355.15 | $(-\mathbf{1.6}\%)$ |
| | 30 | 401.95 | 376.98 | 390.43 | 299.69 | 316.55 | $(5.6\%)$ |
| Marseille | 3 | 466.65 | 469.52 | 487.0 | 498.72 | 476.58 | $(2.1\%)$ |
| | 6 | 285.44 | 288.36 | 290.66 | 297.65 | 289.02 | $(1.3\%)$ |
| | 12 | 206.47 | 205.69 | 207.47 | 204.04 | 195.09 | $(-\mathbf{4.4}\%)$ |
| | 20 | 176.95 | 177.17 | 185.02 | 172.65 | 159.62 | $(-\mathbf{7.5}\%)$ |
| | 30 | 164.84 | 164.84 | 168.23 | 156.09 | 148.71 | $(-\mathbf{4.7}\%)$ |
| Paris | 3 | 185.43 | 187.54 | 189.15 | 176.03 | 192.38 | $(9.3\%)$ |
| | 6 | 118.78 | 115.04 | 124.22 | 106.27 | 116.63 | $(9.7\%)$ |
| | 12 | 82.37 | 85.32 | 87.67 | 74.57 | 81.64 | $(9.5\%)$ |
| | 20 | 82.51 | 79.17 | 82.08 | 64.06 | 67.09 | $(4.7\%)$ |
| | 30 | 76.55 | 81.18 | 76.12 | 61.8 | 61.8 | $(\mathbf{0.0}\%)$ |

## C.4 Example Districting Solutions

We illustrate the districting strategies of the four methods considered in this paper by showing examples of districting solutions on the test instances. Figure 8 shows the districting solutions of the four methods for the city of London with $t = 6$ and $t = 20$ BUs. Figure 9 shows the same results for the city of Manchester with $t = 6$. These solutions correspond to the costs shown in Table 7. These figures show that DISTRICTNET tends to return compact, homogeneous districts. On the contrary, the three benchmarks tend to find districts in a "U" shape. This is especially visible for large district sizes such as $t = 20$. Interestingly, BD and FIG provide visually similar results, especially for the city of London.

Table 8: Compactness of districts found by the different methods. A higher value indicates better compactness.

| City | BD | FIG | PREDGNN | AVGTSP | DISTRICTNET |
|------|------|------|---------|--------|-------------|
| Bristol | 0.263 | 0.260 | 0.307 | 0.293 | **0.375** |
| Leeds | 0.266 | 0.259 | 0.279 | 0.328 | **0.393** |
| London | 0.256 | 0.246 | 0.213 | 0.305 | **0.375** |
| Lyon | 0.298 | 0.296 | 0.287 | 0.357 | **0.370** |
| Manchester | 0.253 | 0.276 | 0.223 | 0.308 | **0.351** |
| Marseille | 0.222 | 0.224 | 0.260 | 0.278 | **0.290** |
| Paris | 0.223 | 0.212 | 0.175 | 0.284 | **0.327** |
| Average | 0.254 | 0.253 | 0.249 | 0.307 | **0.354** |

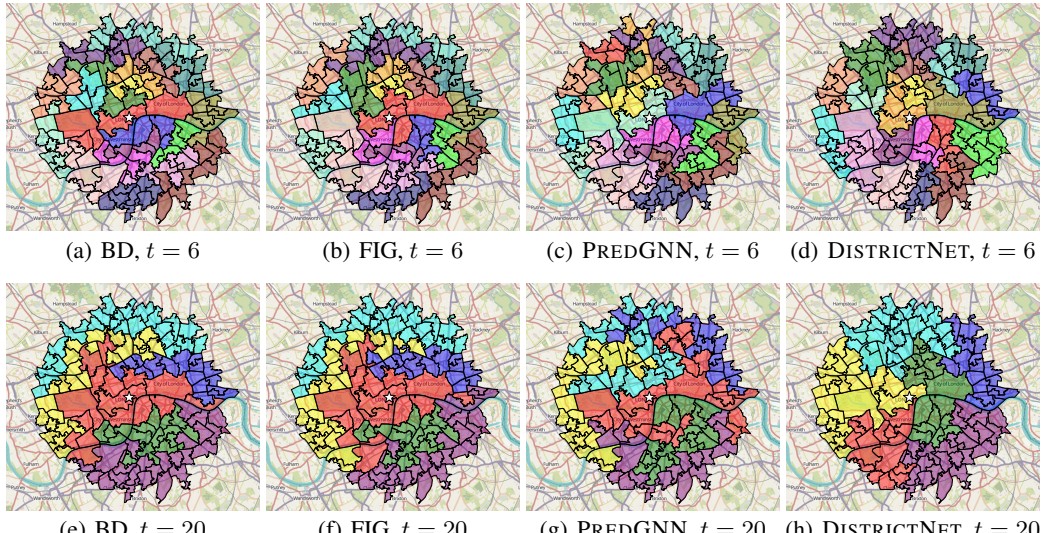

(a) BD, $t = 6$     (b) FIG, $t = 6$     (c) PREDGNN, $t = 6$     (d) DISTRICTNET, $t = 6$

(e) BD, $t = 20$     (f) FIG, $t = 20$     (g) PREDGNN, $t = 20$     (h) DISTRICTNET, $t = 20$

Figure 8: Districting solutions given by BD, FIG, PredictGNN, and DistrictNet for the city of **London** with district target sizes of 6 and 20 BUs. The depot is shown as a white star.

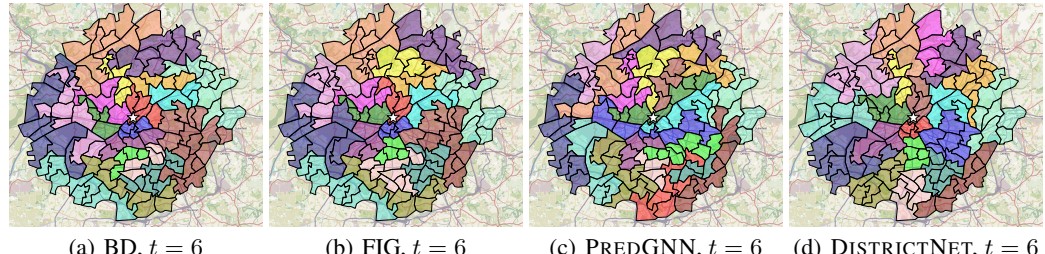

(a) BD, $t = 6$     (b) FIG, $t = 6$     (c) PREDGNN, $t = 6$     (d) DISTRICTNET, $t = 6$

Figure 9: Districting solutions given by BD, FIG, PredictGNN, and DistrictNet for the city of **Manchester** with district target sizes of 6 BUs. The depot is shown as a white star.

