# OpenReview forum: "DistrictNet: Decision-aware learning for geographical districting"
_NeurIPS.cc/2024/Conference — NeurIPS 2024 poster_

### Official Review · Reviewer_VFD5 · 2024-06-27

**Soundness:** 3
**Presentation:** 4
**Contribution:** 3
**Rating:** 6
**Confidence:** 4

**Summary:**

This paper propose an ML-based method to solve districting problems that are known to be challenging. The core idea is to first convert the districting problem to a simpler capacitated minimum spanning tree problem and then leverage the existing learning to optimize method to perform imitation learning. Such conversion is valuable because the learning to optimize approach requires repeatedly solving the optimization problem of interest. Extensive numerical studies were presented, leading to a rich set of insights. I found the paper inspiring.

**Strengths:**

1. Districting problems are common in practice, so I believe this paper is interesting to a broad community.

2. The mapping from a districting problem to a minimum spanning tree problem is elegant and well justified.

3. The numerical studies were well designed and excuted. I particularly appreciate the analysis related to the model's generalizablity, which we do not see in every learning to optimize paper.

4. The paper was well structed and presented. I found it easy to follow.

**Weaknesses:**

- The paper focus on a specific problem class (that is of common interest). The paper can benefit from drawing connections to other problem classes. The idea of converting a harder problem to an easier problem to enable the application of learning to optimize techniques is general and has potential. And I believe there should exist many other examples, in transportation and beyond. Conducting additional experiments on other problems is probably out of the scope of this paper, but it'd be great if the author can add more discussions related to its broader applicability.

- Discussion/comparison related to some benchmarks from the OR/optimization community are missing. For example, when the demands are uniformly distributed, I believe the second-stage TSP objective value has some very nice closed-form approximations (see literature related to continuous approximation). Such approximation usually leads to tractable districting problem with good performance. I'd be good to see some comparisons related to that.

- I'm not fully convinced about the generalizability from small instances to larger instances as the size range experimented in this paper is relatively narrow.

**Questions:**

- It is unclear to me what the costs (objective) are in the numerical experiments.

- Related to weakness 1, what is the downside of performing such conversion?

**Limitations:**

Discussed.

---

> ### Author Rebuttal · Authors · 2024-08-06
>
> Thank you very much for your in-depth review, valuable suggestions, and overall positive appreciation of the paper.
>
> > Some benchmarks from the OR/optimization community are missing. When the demands are uniformly distributed, I believe the second-stage TSP objective value has some very nice closed-form approximations (continuous approximation). Such approximation usually leads to tractable problems with good performance. It'd be good to see some comparisons.
>
> The two benchmarks we call BD and FIG belong precisely to the long stream of research on continuous approximations of TSPs. They trace back to Beardwood et al. (1956): *The shortest path through many points*. Several recent works have shown that the formula of Beardwood et al. (and its extension to districting problems, i.e., BD and FIG) holds remarkably well empirically against more sophisticated regression functions for uniform distributions; see e.g., Kou et al. (2022), *Optimal TSP tour length estimation using standard deviation as a predictor*.
>
> We briefly discussed these results in the introduction of the paper. We will improve the presentation and recall it when introducing the benchmarks in the experiment section.
>
> Note also that we introduced a new benchmark method following the recommendation of Reviewer CwjD, which can be seen as a point-forecast approximation of the stochastic TSP.
>
> > I'm not fully convinced about the generalizability from small instances to larger instances as the size range experimented in this paper is relatively narrow.
>
> In the paper, we train on instances containing 30 basic units (BUs) and test on instances with up to 900 BUs. This is an increase of 30x from train to test, and 7.5x compared to the largest instances of Ferraz et al. This increase is not linear in terms of complexity: the complexity of districting problems increases exponentially with the number of BUs and the size of a district.
>
> Our experiments show that we can scale to the largest and most dense European urban areas and provide good solution qualities. We intentionally train on cities with 30 BUs to show that DistrictNet can learn from small cities and generalize to very large ones. Following your suggestion, we investigate whether our approach can scale even further by considering 2 000 BUs of the Ile-de-France region. The results (see below) show that DistrictNet still performs best. This experiment will be added to the paper.
>
> |   |   BD  |   FIG  |   PredGNN  |   AvgTSP  |   DistrictNet  |
> |---|---|---|---|---|---|
> | Districting cost  | 2379.0  |  2388.8  |  2295.2   |  2262.7  |  **2205.7** |
> | Relative to best | + 7.8\%  | + 8.3\%  | + 4.0\%  | + 2.6\%  |   **0.0**  |
>
> > It is unclear to me what the costs (objective) are in the numerical experiments.
>
> Essentially, we want to find the solution to $\min_{\lambda \in \Lambda} \sum_{d \in \mathcal{D}} C_{TSP}(d) \lambda_d$ where $C_{TSP}(d)$ is the cost of district $d$ computed as the expected cost of a stochastic TSP in district $d$. The true cost $\sum_{d \in \mathcal{D}} C_{TSP}(d) \lambda_d$ is our main performance metric.
>
> All the benchmarks approximate the cost $C_{TSP}(d)$, which is too computationally expensive to calculate in a search algorithm. To do this, they train regression models using features of the districts. DistrictNet does not approximate the districting costs: it learns to parameterize a surrogate optimization problem.
>
> Although all methods have a different approximation for district costs, they are all evaluated using the true performance metric $C_{TSP}(d)$ in our experiments. We will clarify this in the paper, thank you for highlighting this potential for improvement!
>
> > The idea of converting a harder problem to an easier problem to enable the application of learning to optimize techniques is general and has potential. [...] Conducting additional experiments on other problems is probably out of the scope of this paper, but it'd be great if the author can add more discussions related to its broader applicability. [...] What is the downside of performing such conversion?
>
> "Approximating a harder problem by an easier one" can be seen as a paradigm shift for solving difficult combinatorial optimization problems. Classic approaches in OR rely on a purely combinatorial optimization approach. They design an algorithm that works for any problem instance and they tend to focus on worst-case complexity.
>
> On the contrary, the "approximate by an easier problem" literature relies on learning. Its key advantage is being very efficient online because it shifts most of the computing time offline to the training algorithm. Provided that the learning architecture and surrogate optimization layer are well chosen, it enables excellent performance on the test instances if they remain close to the training distribution. Being a discriminative approach (instead of a generative one), it tends to require less data to obtain good performance.
>
> There are also downsides. First, until now, there are no known "worst-case" theoretical guarantees on the quality of the solution. Instead, guarantees are in expectation over the training set, as is usual in ML. Further, the architecture (i.e., the neural network and surrogate optimization layer) may be poorly chosen. Finally, learning may be intensive computationally, both when generating the training instances and in the training algorithm. That is why we study the strategy of training on smaller instances and evaluating on larger ones.
>
> For a general discussion of these aspects and an analytical characterization of generalization bounds, we recommend the preprint that appeared online a few days ago: Aubin-Frankowski et al. (2024), *Generalization Bounds of Surrogate Policies for Combinatorial Optimization Problems*, arXiv:2407.17200.
>
> While an extended discussion is beyond the scope of this paper, we will update our paper to present these advantages and limitations. Thank you for this encouraging suggestion!

---

> > ### Comment · Reviewer_VFD5 · 2024-08-07
> >
> > Thank you for your response. All my concerns have been addressed.

---

> ### Comment · Reviewer_S4NW · 2024-08-11
> **Thank you for your updates**
>
> Dear authors, thank you for your responses. My concerns have been addressed, and the answers clarified my understanding. I'll update my score.

---

> > ### Author Response · Authors · 2024-08-12
> >
> > Thank you for acknowledging our rebuttal and raising your score! We are glad that our rebuttal addressed your concerns.

---

### Official Review · Reviewer_S4NW · 2024-07-12

**Soundness:** 4
**Presentation:** 3
**Contribution:** 3
**Rating:** 6
**Confidence:** 4

**Summary:**

This paper tackles the problem of geographical districting through decision-aware learning. The problem is challenging due to the large combinatorial number of possible district designs. By generalizing the GNN-based method (similar to [Ferraz et al.2024 arXiv]) based on decision-aware learning through the Fenchel-Young loss, the proposed method showed much better relative performance to the existing methods.

**Strengths:**

- The connection between CMST and the districting problem is utilized for decision-aware learning.
- Compared with the recent existing work [Ferraz et al. 2024], a more powerful learning methodology through GNN and FY loss was studied and evaluated, demonstrated using real data (i.e., real districts in famous cities).
- The experimental results show better results than those of existing solvers.

**Weaknesses:**

- Some unclear explanations of using GNN, compared with the existing work [Ferraz et al. 2024]; some parts just follow [Ferraz et al. 2024]. These points raise the insufficient explanation of the technical contributions.
- Datasets are not explained semantically (i.e., geographical differences, why these cities are selected, etc.) in the main text: It is important to explain the importance of the task and used datasets (I understand the page limit and appendix).

**Questions:**

- I am not sure why we should evaluate the p-value in this setting. Where does the randomness come from?
- I understood that the average demands are required. Once BU and average demands are fixed, the solution seems to be (almost) unique. Can we combine some learning-based demand predictors? If the estimation is not correct, how can the districting results be affected? Some questions could be about district costs.

~~~

After discussions, I have updated my score.

**Limitations:**

In my opinion, the authors have addressed the issue adequately and mentioned these points.

---

> ### Author Rebuttal · Authors · 2024-08-06
>
> We thank you very much for your review and detailed evaluation of our paper.
>
> > Some unclear explanations of using GNN, compared with the existing work [Ferraz et al. 2024]; some parts just follow [Ferraz et al. 2024]. These points raise the insufficient explanation of the technical contributions.
>
> Thank you for this remark. We follow Ferraz et al. (2024) in the sense that we use a GNN to embed graphs in a latent space. However, there is a major distinction: we use the GNN to parameterize a surrogate optimization model rather than to estimate the cost of a district. This is not a direct application at all. It requires a suitable loss function, constructing CMST targets to learn from, and an appropriate method to backpropagate gradients. The major benefit of this approach is its ability to generalize. As shown in Table 1, it is the combination of GNN and the CMST layer that allows finding good solutions with a single model on all the cities and sizes used in our experiments.
>
> > Datasets are not explained semantically (i.e., geographical differences, why these cities are selected, etc.) in the main text: It is important to explain the importance of the task and used datasets (I understand the page limit and appendix).
>
> Thank you for this suggestion. We agree that the presentation of the datasets can be improved. To provide comparable results with Ferraz et al, we include the same districting tasks on cities in England. However, we expand the experimental setting by considering larger cities and by including cities in France. Thus, we consider some of the largest and most dense European urban areas. We show that we can solve real-world districting tasks and generalize over cities that are diverse in terms of geography and social characteristics.
>
> > I am not sure why we should evaluate the p-value in this setting. Where does the randomness come from?
>
> In Table 1, we measure the empirical performance of our method on 35 districting tasks from real-world cities. The randomness comes from the selection of the test instances from the distribution of possible districting tasks. The $p$-value tells us that, even though we have a limited set of tasks, we can confidently tell that the difference in average performance is not due to random chance but to a clear advantage in the method.
>
> > I understood that the average demands are required. Once BU and average demands are fixed, the solution seems to be (almost) unique. Can we combine some learning-based demand predictors? If the estimation is not correct, how can the districting results be affected? Some questions could be about district costs.
>
> Formally, we solve the problem
>
> $\min_{\lambda \in \Lambda}\sum_{d \in \mathcal{D}} E_{\xi}[c_{TSP}(d, \xi)] \lambda_d$,
>
> where $c_{TSP}(d, \xi)$ is the cost of the TSP in district $d$ with requests $\xi$. The suggestion of the reviewer is to solve instead
>
> $\min_{\lambda \in \Lambda} \sum_{d \in \mathcal{D}} c_{TSP}(d, E_\xi[\xi]) \lambda_d$,
>
> i.e., switching the expectation and minimization operator.
>
> We did not implement this benchmark because this approximation leads to poor results in many stochastic problems (see e.g., Chapter 4.2 of Birge and Louveaux, *Introduction to stochastic programming*). This is because this approximation ignores the variance of the random variable $\xi$.
>
> We followed your suggestion and implemented this method as a baseline. In our setting, the demand distribution is known, so we do not need to forecast it. For each basic unit, the average demand request $E_\xi[\xi]$ is a single request that appears at the barycentre of the area. Evaluating a district cost reduces to solving a TSP over the barycentres of all its BUs.
>
> We integrate this approximation in an iterated local search and evaluate all our districting tasks. The results (called AvgTSP in the table below) show that this method finds districts of reasonably high quality and outperforms the other baselines. Still, DistrictNet outperforms it by more than 4\% on average.
>
> | Method        | Avg Rel Diff       | p-value              |
> |---------------|--------------------|----------------------|
> | Benchmark 1, BD    | 10.02 \% | 1.3e-08 |
> | Benchmark 2, FIG   | 10.17 \% | 1.6e-08  |
> | Benchmark 3, PredGnn  | 11.56 \% | 1.5e-10  |
> | **Benchmark 4, AvgTSP**   | 4.44 \% | 2.7e-04  |
> | DistrictNet   | **0.0 \%**             |     -                |
>
> We will add this method to the paper with a formal description and a detailed analysis of its performance. Thank you for this great suggestion!

---

### Official Review · Reviewer_WW6p · 2024-07-12

**Soundness:** 2
**Presentation:** 2
**Contribution:** 2
**Rating:** 6
**Confidence:** 1

**Summary:**

The paper discusses the development and evaluation of a new method called
DISTRICTNET for addressing districting and routing problems using neural networks.
The approach focuses on minimizing the Fenchel Young loss and generalizing to large
out-of-distribution instances from training on smaller instances. The method involves
a representation of the instances as a set of graph weights learned by GNN. These weights
are fed to a CMST algorithm to obtain the spatial partitioning. Comparative
benchmarks include linear regression models and graph neural networks trained
for cost estimation. Results show that DISTRICTNET can achieve lower costs with
a small number of training examples but benefits from more examples,
with diminishing returns.

**Strengths:**

Some key strengths of the research presented in this paper include a structured learning approach to obtain high-quality solutions to large districting problems, the robustness of the approach to changes in problem parameters, and the ability to generalize to out-of-distribution instances.

**Weaknesses:**

I am not familiar with the routing literature and I had a very hard time to read this paper. The supplementary
material is essential and the authors should consider if the mathematical treatment
of several issues that were not relevant for the algorithm proposed should not be moved to
the appendix in exchange of bringing some of the appendix to the main paper. For example,
I was lost about the district demand and cost features along the paper. This is
explained in the appendix.

The datasets are small, with the number of areas varying between 120 and 983.
They use an additional set of 27 cities, 25 of them with less than 100 areas and a
maxium of 684.

The benchmarks did not include what the authors call the quick heuristic methods.
I also found other recent papers in important venues dealing with this districting
problem by partitioning a minimum spanning tree as in the paper and with simpler
methods.
Teixeira, L. V., Assunção, R. M., & Loschi, R. H. (2019). Bayesian space-time partitioning by sampling and pruning spanning trees. Journal of Machine Learning Research, 20(85), 1-35.
Luo, Z. T., Sang, H., & Mallick, B. (2021). A Bayesian contiguous partitioning method for learning clustered latent variables. Journal of Machine Learning Research, 22(37), 1-52.
Luo, Z. T., Sang, H., & Mallick, B. (2021). BAST: Bayesian additive regression spanning trees for complex constrained domain. Advances in Neural Information Processing Systems, 34, 90-102.
McCartan, C., & Imai, K. (2023). Sequential Monte Carlo for sampling balanced and compact redistricting plans. The Annals of Applied Statistics, 17(4), 3300-3323.

**Questions:**

Add "routing" to the paper title.

Each benchmark method has a different definition of a district cost. The outputs of each of
these algorithms are a result of these different cost definitions or are due to their
different approaches?

**Limitations:**

This is a theoretical paper with no immediate connection to societal impact.

---

> ### Author Rebuttal · Authors · 2024-08-06
>
> Thank you very much for your detailed review of the paper and the helpful suggestions. We respond to your concerns and questions below.
>
> > The supplementary material is essential and the authors should consider [...] bringing some of the appendix to the main paper. For example, I was lost about the district demand and cost features along the paper.[...] Each benchmark method has a different definition of a district cost. The outputs of each of these algorithms are a result of these different cost definitions or are due to their different approaches?
>
> Thank you for this comment. Essentially, we want find a districting solution $\lambda$ that minimizes $\min_{\lambda \in \Lambda} \sum_{d \in \mathcal{D}} C_{TSP}(d) \lambda_d$, where $C_{TSP}(d)$ is the cost of district $d$. A district's cost is the expected cost of a "stochastic" TSP. The true cost $\sum_{d \in \mathcal{D}} C_{TSP}(d) \lambda_d$ is the main performance metric. This is the cost that we measure in all our experiments.
>
> Evaluating a district's cost $C_{TSP}(d)$ is too computationally demanding to be integrated into a districting algorithm at all steps of the search. Hence, the benchmarks approximate this cost during the search. This will be clarified in the paper.
>
> To approximate districts' costs, the benchmarks train regression models using features of the districts. In contrast, DistrictNet does not learn to approximate the districting costs: it learns to parameterize a surrogate optimization problem. In the final version, we will present the key district features in the main body of the paper. Thank you for this suggestion!
>
> > The datasets are small, with the number of areas varying between 120 and 983.
>
> In the paper, we train on instances containing 30 basic units (BUs) and evaluate on instances with up to 900 BUs. This is an increase of 30x from train to test, and 7.5x compared to the largest instances of Ferraz et al. (2024). This increase is not linear in terms of complexity: the complexity of districting problems increases exponentially with the number of BUs and the size of a district.
>
> Our experiments show that we can scale to the largest and most dense European urban areas and provide good solution qualities. Following your suggestion, we investigate whether our approach can scale even further by considering 2 000 BUs of the Ile-de-France region. DistrictNet provides the best performance, showing that it generalizes to instances that are more than 60 times larger than the training ones. This experiment will be added to the paper.
>
> |   BD  |   FIG  |   PredGNN  |   AvgTSP  |   DistrictNet  |
> |----|----|----|----|----|
> | 2379.0  |  2388.8  |  2295.2   |  2262.7  |  **2205.7** |
> | + 7.8\%  | + 8.3\%  | + 4.0\%  | + 2.6\%  |   **0.0**  |
>
> > They use an additional set of 27 cities, 25 of them with less than 100 areas.
>
> We want to clarify: the 27 small cities presented in Appendix B.2 are used to generate the training data. Our test instances are at least four times larger containing more than 120 BUs. We intentionally train on cities with 30 BUs to show that DistrictNet can learn from small cities and generalize to much larger ones.
>
> > The benchmarks did not include what the authors call the quick heuristic methods.
>
> We do include two quick heuristic methods: BD and FIG, which apply an iterated local search with continuous approximations of district costs. These methods stem from the seminal work of Beardwood et al. (1956): \textit{The shortest path through many points}. Several recent works have shown that the formula of Beardwood et al. holds remarkably well against more sophisticated regression functions for uniform distributions; see, e.g., Kou et al. (2022), *Optimal TSP tour length estimation using standard deviation as a predictor*.
>
> Further, following the recommendation of Reviewer CwjD, we introduced a new benchmark method. It approximates a district's cost by the cost of the TSP that goes through the barycentre of each of its BUs.
>
> > I also found other recent papers in important venues dealing with this districting problem by partitioning a minimum spanning tree as in the paper and with simpler methods.
>
> Thank you for referring us to the rich literature on partition sampling. The suggested works provide methods to sample districts with constraints on the balance, contiguity, and compactness of the partitioning. Balance is to be understood w.r.t. features such as mortality rates in Teixeira et al. These approaches are motivated by the fact that sampling from a uniform distribution of partitions tends to have low compactness. Hence, they recommend sampling from the spanning tree distribution.
>
> Our approach is fundamentally different. It is discriminative, in the sense that it strives to identify the single partition with minimum cost. In contrast, the suggested papers are generative: they output a distribution of partitions. Further, none of them considers districting-and-routing applications, and adapting them to handle routing applications is not straightforward.
>
> Still, our work and the literature on partition sampling have very interesting connections. A first idea could be to solve approximately the CMST problems by sampling from balanced partitioning partitions. In this way, we might replace our current search algorithm (iterated local search) with a specialized type of random search over spanning trees. A second idea is to integrate considerations about balance and compactness into the problem. To this end, given examples of districts that are compact and balanced, DistrictNet can be directly trained to learn to imitate them using a parameterized CMST.
>
> We will include this discussion in the paper, and highlight the opportunities in future research on the integration of decision-aware partitioning and sampling from spanning trees. Thank you for this suggestion!
>
> > Add "routing" to the paper title.
>
> Yes, we will specify that we solve districting-and-routing problems in the paper title.

---

> > ### Comment · Area_Chair_Jr4b · 2024-08-12
> > **Reviewers WW6p and qHFv**
> >
> > The authors have provided extensive replies to the criticisms expressed in your reviews. Have these responses satisfied your concerns? If not, are there any further clarifying questions you would like to ask? The author discussion period ends tomorrow (Aug 13). Please respond.

---

> > ### Comment · Reviewer_WW6p · 2024-08-13
> >
> > Thanks for addressing my questions and providing more detail. I will update my score.

---

> > > ### Author Response · Authors · 2024-08-13
> > >
> > > Thank you for your response and raising your score!

---

### Official Review · Reviewer_qHFv · 2024-07-13

**Soundness:** 3
**Presentation:** 3
**Contribution:** 3
**Rating:** 5
**Confidence:** 4

**Summary:**

The authors use CMST to solve the problem of districting and routing in large scale scenarios. Finding the relationship between CMST and partitioning is quite beneficial for researchers engaged in related research.

However, it is worth noting that the authors only report the performance of the model on the 'cost' metric, and do not investigate the performance of the method on the common metrics of traditional districting task. What's more, they do not discuss the efficiency of the method. I hope that the authors can supplement this.

Another problem is that the baseline methods compared seem to be relatively old except for PREDGNN. The districting methods I know in terms of road network, such as CCH, have not been used as the baseline method. I hope the author can clarify the criteria for baseline selection.

**Strengths:**

The authors use CMST to solve the problem of districting and routing in large scale scenarios. Finding the relationship between CMST and partitioning is quite beneficial for researchers engaged in related research.

**Weaknesses:**

It is worth noting that the authors only report the performance of the model on the 'cost' metric, and do not investigate the performance of the method on the common metrics of traditional districting task. What's more, they do not discuss the efficiency of the method. I hope that the authors can supplement this.

Another problem is that the baseline methods compared seem to be relatively old except for PREDGNN. The districting methods I know in terms of road network, such as CCH, have not been used as the baseline method. I hope the author can clarify the criteria for baseline selection.

**Questions:**

Please see the weaknesses.

**Limitations:**

Yes. The authors claim that though the proposed method can be applied to a wide variety of partitioning problems, they only focus on the districting and routing problem in this paper. In fact, I think it's a very valuable question to discuss because the evaluation metric of the partitioning problem and that of the districting and routing problem are different. I think it is necessary for the authors to give more explanation of why their method can be applied to a wider range of partitioning problems.

---

> ### Author Rebuttal · Authors · 2024-08-06
>
> Thank you for your review and insightful comments. We answer your questions below and clarify the concerns discussed in the report.
>
> > The authors only report the performance of the model on the 'cost' metric, and do not investigate the performance of the method on the common metrics of traditional districting task. [...] The authors claim that though the proposed method can be applied to a wide variety of partitioning problems, they only focus on the districting and routing problem in this paper. [...] I think it is necessary for the authors to give more explanation of why their method can be applied to a wider range of partitioning problems.
>
> Thank you for your comments. In this work, we focus on the 'cost' metric because our task at hand is to solve the problem $\min_{\lambda \in \Lambda} \sum_{d \in \mathcal{D}} C_{TSP}(d) \lambda_d$, where $C_{TSP}(d)$ is the expected cost of a stochastic TSP in district $d$. In other words, we want to partition the city into efficient districts from a logistics perspective. The cost of a district corresponds to the long-term expected cost of deliveries (TSP) in each area. In contrast with other districting objectives for which a closed-form formula is available (e.g., compactness) this setting is computationally challenging. This is because closed-form formulas to approximate TSP costs are inaccurate, and sampling scenarios and solving TSPs is too time-consuming to be integrated in a districting algorithm.
>
> Our contribution is a new decision-aware approximation and solution method. Our approach can be readily extended to other districting metrics. This is a valuable direction for future work, as also highlighted by Reviewer WW6p. To adapt DistrictNet to this setting, we only need to adapt the generation of the training instances. This is why we claim that our framework applies to a wide range of constrained partitioning problems: it can work with any general districting cost function of the form $C(d)$. Hence, it can consider other metrics such as fairness, balancing, compactness, etc. Because the task at hand remains a constrained partitioning problem, our surrogate CMST layer will still capture the structure of the problem.
>
> We will make sure to include additional discussions on this aspect in the final version of the paper. We will suggest exploring other districting metrics as future work, and specify why DistrictNet is especially suitable in these settings.
>
> Lastly, we have evaluated the compactness of our districting solutions using Reock's formula. The results (see below; a larger value indicates higher compactness) demonstrate that DistrictNet generally achieves the highest compactness on average. However, it is crucial to note that, in logistics, delivery costs are not always correlated with compactness. The shape and orientation of the districts relative to the depot's position can significantly impact TSP costs.
>
> |    | BD  | FIG | PredGNN | AvgTSP | DistrictNet |
> |---|---|---|---|---|---|
> | Bristol |0.124 | 0.125 | 0.124 | 0.142 | **0.173** |
> | Leeds |0.168 | 0.154 | 0.187 | 0.223 | **0.268** |
> | London |0.086 | 0.089 | 0.08 | 0.096 | **0.11** |
> | Lyon |0.183 | 0.177 | 0.209 | 0.213 | **0.258**|
> | Manchester |0.167 | 0.165 | 0.142 | 0.258 | **0.272** |
> | Marseille |0.143 | 0.143 | 0.177 | **0.195** | 0.185 |
> | Paris |0.169 | 0.182 | 0.157 | **0.278** | 0.274 |
> | *Average* |0.149 | 0.148 | 0.154 | 0.201 | **0.22** |
>
> > The authors do not discuss the efficiency of the method. I hope that the authors can supplement this.
>
> Our algorithm is made of three main components: the data generation, the learning algorithm (setting the weights of the neural network), and the inference algorithm (using the neural network to obtain a solution on a given instance). Learning is quite fast (see Table 2 in the appendix) and inference is very fast (20 minutes). For DistrictNet, the crux of computations is in the generation of training data. Yet, we show in Figure 5 that DistrictNet provides already good results with only 20 training examples.
>
> We will improve the discussion of the efficiency of our methods following your recommendation.
>
> > Another problem is that the baseline methods compared seem to be relatively old except for PREDGNN. [...] I hope the author can clarify the criteria for baseline selection.
>
> To our knowledge, the recent work of Ferraz et al. (2024) represents the state-of-the-art for districting and routing problems. The two benchmarks BD and FIG stem from the long history of analytical results on TSPs and, in particular, the formula of Beardwood et al. (1959) on approximation formulas in asymptotic regimes with many points. The recent work of Kou et al. (2022), *Optimal TSP tour length estimation using standard deviation as a predictor*, shows that more sophisticated regression methods do not significantly outperform the formula of Beardwood et al. That is why we include the two benchmarks BD and FIG even if they are relatively older.
>
> This discussion will be added to the paper. Note also that we have included an additional benchmark method (AvgTSP) following the suggestion of Reviewer S4NW.
>
> > The districting methods I know in terms of road network, such as CCH, have not been used as the baseline method.
>
> Thank you for this suggestion. We understand that you refer to Customizable Contraction Hierarchies (CCH), an efficient technique to compute distances over road networks. Please correct us if our interpretation is wrong.
>
> Since CCH is a technique designed to speed up shortest-path computations, it cannot be directly used in our districting-and-routing context to contract edges and speed up the method. This is because (1) customer positions in each district are random (they are calculated over 100 scenarios), and (2) we calculate TSPs and not shortest paths.
>
> We thank you again for your review, which allowed us to significantly improve the discussion of the general value of DistrictNet for districting.

---

### Author Rebuttal · Authors · 2024-08-06

## We thank the reviewers for their constructive comments and feedback

Dear Reviewers, Dear Area Chair,

We want to express our sincere thanks for the detailed reviews of our work and the constructive feedback. The reviewers have appreciated the value of our decision-aware approach to solve large districting problems and the potential of identifying the CMST as a differentiable optimization surrogate for partitioning problems

Several comments have been raised by multiple reviewers, addressing: (a) the definition of districting costs and the approximations used in the benchmarks, (b) what makes a districting task large, (c) what justifies the choice of benchmarks and whether others could be included, and (d) what makes DistrictNet applicable to other partitioning problems. We have answered these points in detail in each reviewer's rebuttal. We also provide additional results: a new benchmark method following the suggestion of Reviewer CwjD, and results on an even larger districting task with 2 000 basic units.

We are available to answer any remaining questions during the discussion period.

---

### Decision · Program_Chairs · 2024-09-25

**Decision:**

Accept (poster)

**Comment:**

This paper introduces and evaluates a new learning to optimize approach to solving districting problems. The paper's strengths include the following: it focuses on a class of problems that is common in practice and should be of interest to a broad audience; the paper's approach identifies and exploits a novel relationship between CMST and partitioning; and an experimental analysis on real-world data sets show that the approach (1) outperforms a range of benchmark approaches representative of various threads of prior work on this problem (including a additional benchmark approach proposed by one of the reviewers and added during the review period), and (2) shows great ability to generalize to large-scale problems. The major weakness of the paper is that the presentation is confusing in many spots. It is unclear in its explanations of some key technical aspects of the approach and, in other places, assumes background information that can only be found in the supplementary material, all of which leads the reader in doubt abut the significance of the paper's contributions. The reviewers have made several suggestions for how to improve the presentation, and in the discussion period, the authors proposed revisions that make the paper's technical contributions clearer and have satisfied reviewer concerns. Please incorporate all of your proposed additions and revisions in the final version of the paper.